# Less is More: Unlocking Specialization of Time Series Foundation Models via Structured Pruning

**Lifan Zhao**[1]*, **Yanyan Shen**[1]†, **Zhaoyang Liu**[2], **Xue Wang**[2], **Jiaji Deng**[2],
[1]Shanghai Jiao Tong University, [2]Alibaba Group
{mogician233,shenyy}@sjtu.edu.cn, {jingmu.lzy,xue.w,dengjiaji.djj}@alibaba-inc.com

## Abstract

Scaling laws motivate the development of Time Series Foundation Models (TSFMs) that pre-train vast parameters and achieve remarkable zero-shot forecasting performance. Surprisingly, even after fine-tuning, TSFMs cannot consistently outperform smaller, specialized models trained on full-shot downstream data. A key question is how to realize effective adaptation of TSFMs for a target forecasting task. Through empirical studies on various TSFMs, the pre-trained models often exhibit inherent sparsity and redundancy in computation, suggesting that TSFMs have learned to activate task-relevant network substructures to accommodate diverse forecasting tasks. To preserve this valuable prior knowledge, we propose a structured pruning method to regularize the subsequent fine-tuning process by focusing it on a more relevant and compact parameter space. Extensive experiments on seven TSFMs and six benchmarks demonstrate that fine-tuning a smaller, pruned TSFM significantly improves forecasting performance compared to fine-tuning original models. This "prune-then-finetune" paradigm often enables TSFMs to achieve state-of-the-art performance and surpass strong specialized baselines. Source code is made publicly available at `https://github.com/SJTU-DMTai/Prune-then-Finetune`.

## 1 Introduction

Time Series Foundation Models (TSFMs) are large forecasting models pre-trained on massive datasets. Recent researches on TSFMs [1, 5, 11, 12, 21, 27, 30] have demonstrated their impressive zero-shot forecasting capabilities across various domains without being specifically trained for those particular prediction tasks. Time series forecasting has a distinct advantage compared to other machine learning tasks. That is, new data naturally accumulates over time, providing fresh training samples without manual labeling. In many industrial sectors such as traffic monitoring, energy consumption, weather prediction, and financial analysis, organizations continuously collect extensive time series data. This data serves two purposes: it provides a long historical context for zero-shot forecasting and abundant samples for model training. Given this data availability, it becomes crucial to evaluate whether generic TSFMs offer real advantages over specialized models trained on task-specific datasets.

Our performance evaluations in Fig. 1 demonstrate that even state-of-the-art pretrained TSFMs operating in zero-shot mode frequently underperform compared to specialized models like PatchTST [24]. This performance gap primarily stems from statistical differences between the diverse pre-training data and the specific patterns in downstream time series. Although fine-tuning TSFM parameters on target datasets improves prediction accuracy, our experiments show that even these adapted TSFMs struggle to consistently outperform PatchTST across various benchmarks. This raises a crucial research question: how can we more effectively leverage these powerful TSFMs to achieve superior performance in specific forecasting applications?

---

*This work was done during his internship at Alibaba Group.
†corresponding author

39th Conference on Neural Information Processing Systems (NeurIPS 2025).

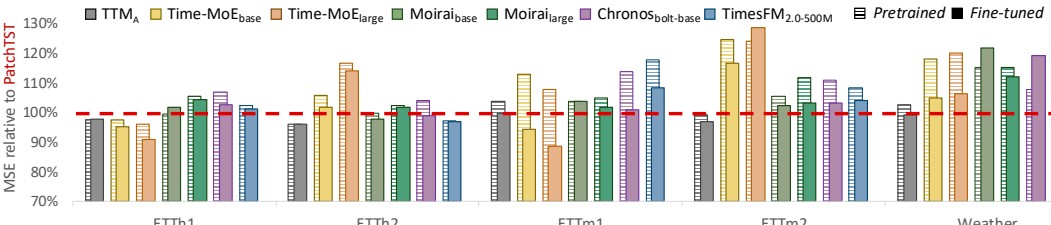

Fig. 1: Performance comparison between TSFMs and PatchTST trained on full-shot training data of each benchmark. We calculate the average MSE of forecasting 96, 192, 336, and 720 steps, which is further divided by that of PatchTST with well-tuned hyperparameters [26]. Weather is used for pre-training TimesFM and is not evaluated. The red dashed line represents the PatchTST baseline.

To answer the question, we investigate the internal characteristics of TSFMs that help explain their behaviors. Our analysis reveals that the pre-trained TSFMs often exhibit inherent sparsity in hidden states and redundancy in parameters. For instance, up to 50% of attention heads yield negligible outputs throughout a downstream task, and up to 60% of intermediate channels in feed-forward networks are never activated. These sparsity patterns vary across datasets, suggesting that TSFMs activate different subnetworks depending on the specific task. While conventional fine-tuning retains the original model architecture, it may disrupt these task-relevant activation patterns by modifying all parameters indiscriminately. This approach risks overfitting and fails to capitalize on the valuable task-specific substructures identified during pre-training.

In this paper, we propose decomposing TSFM deployment into two stages: architectural specialization through *structured pruning* [3, 16] and weight specialization through fine-tuning. While structure pruning by removing redundant components like channels or attention heads is commonly used to compress large models for lower latency and memory usage [22, 29], our primary objective is to facilitate fine-tuning in a narrowed parameter space for stable performance improvement. Specifically, for each downstream task, we analyze channel importance across all linear transformation layers of a TSFM using a loss-guided metric [22] to identify and remove unimportant model parameters. This results in a sparse model whose remaining parameters are fine-tuned on the downstream data.

We conduct extensive experiments on various TSFMs and six popular time series forecasting benchmarks. The results demonstrate that fine-tuning a pruned TSFM yields better forecasting performance than fine-tuning the original model in most cases, achieving up to 22% error reduction. In particular, when selecting the best TSFM on a specific benchmark, our prune-then-finetune approach increases the win rate against PatchTST from 90% to 100%. Moreover, a pruned model achieves superior zero-shot performance when forecasting other time series within the same domain, indicating that the identified model substructure transfers effectively across similar datasets. These findings highlight the essential role of architectural specialization for TSFMs and establish the prune-then-finetune paradigm as a promising strategy for effective TSFM deployment. The major contributions of the paper are summarized in the following.

- **Identifying the TSFM specialization bottleneck:** we provide comprehensive and fair benchmark comparisons showing that even fine-tuned TSFMs struggle to outperform specialized baselines in full-shot forecasting scenarios, highlighting TSFM specialization as a critical unresolved challenge.

- **Uncovering inherent sparsity in TSFMs:** our empirical analysis reveals that pre-trained TSFMs, regardless of whether they utilize sparse Mixture-of-Experts architectures, activate only task-relevant parameter subsets during downstream inference. We argue that these specialized architectural components represent valuable prior knowledge acquired during pre-training and should be preserved when forecasting time series from specific domains.

- **Pioneering pruning for TSFMs:** we propose a novel prune-then-finetune paradigm as an effective post-training pipeline for TSFMs. To the best of our knowledge, this is the first work that explores pruning TSFMs to enhance forecasting performance. Remarkably, we find that up to 90% parameters can be removed while simultaneously improving performance.

- **Demonstrating performance and efficiency gains:** we conduct extensive experiments on various TSFMs and datasets. The results demonstrate that our minimalist pruning approach frequently enables TSFMs to surpass state-of-the-art baselines in full-shot forecasting scenarios. Additionally, our structured pruning technique delivers up to $7\times$ speedup in inference efficiency.

## 2 Related Works

### 2.1 Time Series Foundation Models

Recently, increasing efforts have been devoted to pre-training TSFMs on large-scale time series data for a versatile starting point of various prediction tasks. Timer [21] and Timer-XL [20] [1] stack multiple Transformer layers with causal attention to capture temporal correlations. Furthermore, several specialization modules are designed to handle the heterogeneity of time series. For frequency-specific specialization, Moirai [30] differentiates input/output projection layers among frequencies, while TTM [7] and TimesFM [5] include categorical frequency embeddings as input. Time-MoE [28] and Moirai-MoE [19] adopt a sparse mixture-of-experts (MoE) design where tokens are processed by different expert feed-forward networks (FFNs). Though only a few experts are activated for each token, all experts can be used and fine-tuned on the whole dataset.

While some TSFMs claim state-of-the-art performance, their comparisons set the lookback length to 96 for specialized models (e.g., PatchTST [24]) while thousands for TSFMs. Li et al. demonstrate that specialized models with well-tuned hyperparameters outperform TSFMs, while they did not tune TSFM hyperparameters. In this work, after extensively tuning hyperparameters of all models, our benchmark results confirm that TSFMs still cannot surpass specialized baselines. Also, we find that applying parameter-efficient fine-tuning [13] to TSFMs is not superior to full fine-tuning (see Appendix C.3) and cannot resolve the limitations in specialization. A concurrent work [25] also emphasizes that how to effectively fine-tuning TSFMs is an under-explored yet important problem.

### 2.2 Model Pruning

Pruning [4], a conventional technique for network compression and acceleration, is increasingly applied to Large Language Models (LLMs) for efficient inference. Based on pruning units, pruning has two categories. Unstructured pruning removes unimportant elements of weight matrices, while the sparse matrices require specialized hardware or software optimizations for acceleration [36]. Structured pruning removes entire components (e.g., channels and attention heads with the lowest importance) and is hardware-friendly. Numerous pruning metrics have been proposed to determine whether a pruning unit is essential and should be pruned or not. Various metrics guide pruning; for example, LLM-Pruner [22] assesses a component's importance by its impact on loss. Typically, LLMs after structure pruning struggle to regain original performance even after retraining [2, 22, 36]. The potential reason behind lossy compression of LLMs is that various neurons can preserve language skills and vast factual knowledge, which is critical to commonsense questions.

Crucially, the motivation of our work differs from the predominant use of pruning for LLMs. Our main purpose is not to pursue inference efficiency. Instead, we are devoted to improving the fine-tuning performance of TSFMs by pruning neurons that are less relevant to the downstream task. The rationale is that time series are characterized by strong heterogeneity across domains and frequencies [19], and a specific task may only need to activate a certain fraction of model parameters [28]. Moreover, due to the nature of non-stationarity, time series usually exhibit various data patterns over time, and there are scarce latent features that maintain long-term effectiveness for time series forecasting. Thus, pruning has the potential to focus TSFMs on the most critical channels for a given task.

## 3 Preliminary

In this section, we first formalize the notations of key modules in Transformers, which constitute the backbone of most existing TSFMs. Then, we conduct a preliminary study on TSFMs in aspects of attention head outputs and FFN activations. Additionally, analysis of parameter magnitudes is provided in Appendix B.1.

### 3.1 Transformer Architecture

**Multi-Head Attention (MHA).** Typically, TSFMs organize time series observations as a sequence of tokens (a.k.a., patches). Let $T$ denote the number of input tokens and $\mathbf{X} \in \mathbb{R}^{T \times d}$ denote the token embeddings. Suppose there are $H$ attention heads and each head has $d_H$ dimensions. For each $i \in \{1, \cdots, H\}$, let $\mathbf{W}_i^K, \mathbf{W}_i^Q, \mathbf{W}_i^V \in \mathbb{R}^{d \times d_H}$ denote key, query, value projections for the $i$-th head,

Table 1: Statistics of popular TSFMs and PatchTST. $d_{\text{ffn}}$ denote the intermediate dimension of FFN.

| Model | TTM$_A$ | Time-MoE$_{\text{base}}$ | Time-MoE$_{\text{large}}$ | Moirai$_{\text{base}}$ | Moirai$_{\text{large}}$ | Chronos$_{\text{bolt-base}}$ | TimesFM$_{2.0}$ | PatchTST |
|---|---|---|---|---|---|---|---|---|
| Architecture | MLP | Decoder-Only | Decoder-Only | Encoder-Only | Encoder-Only | Encoder-Decoder | Decoder-Only | Encoder-Only |
| Model Size | 5M | 113M | 453M | 91M | 311M | 205M | 500M | 500K |
| #Layer | 6 | 12 | 12 | 12 | 24 | 24 | 50 | 3 |
| #Head | \ | 12 | 12 | 12 | 16 | 12 | 16 | 12 |
| $d$ | 26∼384 | 384 | 768 | 768 | 1024 | 1024 | 1280 | 128 |
| $d_{\text{ffn}}$ | 52∼768 | 1536 | 3072 | 3072 | 4096 | 3072 | 1280 | 256 |

and $\mathbf{W}_i^O \in \mathbb{R}^{d_H \times d}$ for output projections. Formally, MHA computation can be defined as below:

$$\texttt{MHA}(\mathbf{X}) = \sum_i \mathbf{O}_i = \sum_i \left( \texttt{softmax}\left( \frac{(\mathbf{X}\mathbf{W}_i^Q)(\mathbf{X}\mathbf{W}_i^K)^\top}{\sqrt{d_H}} \right) \mathbf{X}\mathbf{W}_i^V \right) \mathbf{W}_i^O, \tag{1}$$

where $\mathbf{O}_i \in \mathbb{R}^{T \times d}$ is the output of the $i$-th head.

**Feed-Forward Network (FFN).** Each FFN involves at least two linear layers parameterized by $\mathbf{W}^{\texttt{up}}, \mathbf{b}^{\texttt{up}}, \mathbf{W}^{\texttt{down}}, \mathbf{b}^{\texttt{down}}$. FFN computation can be written as

$$\texttt{FFN}(\mathbf{X}) = \mathbf{W}^{\texttt{down}} \underbrace{\sigma(\mathbf{X}\mathbf{W}^{\texttt{up}} + \mathbf{b}^{\texttt{up}})}_{\text{activations}} + \mathbf{b}^{\texttt{down}}, \tag{2}$$

where $\sigma$ is the activation function (e.g., ReLU, GELU, SwiGLU) that can deactivate some intermediate hidden states as zeros or tiny negative values. TTM [7], though not based on Transformers, also incorporates MLPs with GELU activation. For the $i$-th intermediate FFN channel, the probability of producing positive activations over a dataset can reflect its relevance to the downstream task.

**Residual Connection.** To enable deeper architectures and stabilize training, Transformer employs residual connections and normalizations. Formally, the output of a Transformer can be written as

$$\mathbf{X} \leftarrow \mathbf{X} + \texttt{MHA}(\texttt{Norm}(\mathbf{X})), \quad \mathbf{X} \leftarrow \mathbf{X} + \texttt{FFN}(\texttt{Norm}(\mathbf{X})), \tag{3}$$

where Norm can be implemented by layer normalization or RMSNorm.

## 3.2 Empirical Analysis of TSFMs

According to the theory of bias-variance trade-offs [9], a large model can induce high variance risks in predictions, which seems contradictory to the skilled zero-shot performance of TSFMs on unseen tasks. To ascertain the mechanism of TSFMs, we conduct an investigation into TSFMs through the lens of attention heads and activations in FFNs. We perform 96-step forecasting on the Weather [31] and ETTm1 [35] datasets by PatchTST and seven pre-trained TSFMs, including TTM [7], Time-MoE$_{\text{base}}$, Time-MoE$_{\text{large}}$ [28], Moirai$_{\text{base}}$, Moirai$_{\text{large}}$ [30], Chronos-bolt$_{\text{base}}$ [1], and TimesFM-2.0-500M [5]. The key statistics of the TSFMs are provided in Table 1.

### 3.2.1 Insignificant Outputs of Attention Heads

Empirically, we find that there are a few attention heads in TSFMs that usually produce insignificant modifications on residuals. For each head, we calculate its relative output norm averaged over the downstream dataset, as defined below.

**Definition 1** Given the $i$-th head output $\mathbf{o}_i \in \mathbb{R}^d$ and the residual $\mathbf{x} \in \mathbb{R}^d$ of each input token, the *average relative output norm* of the $i$-th head is defined as $\mathbb{E}_{\mathbf{x}}(\|\mathbf{o}_i\|_2 / \|\mathbf{x}\|_2)$ over a downstream dataset, which can reflect the importance of the $i$-th head when modifying the residual.

Fig. 2 illustrates diverse distributions of average relative output norms. Compared to PatchTST where all heads actively contribute $\geq 3\%$ relative norm, a few heads of pre-trained TSFMs give nearly zero norms, and larger TSFM variants tend to exhibit greater sparsity. For instance, 50% (*resp.* 30%) heads of Time-MoE$_{\text{large}}$ (*resp.* Moirai$_{\text{large}}$) produce outputs with negligible relative norms ($< 1\%$), while their smaller counterparts have fewer insignificant heads. Nevertheless, with parameters more than Time-MoE$_{\text{base}}$ and Moirai$_{\text{base}}$, Chronos exhibits lower sparsity, suggesting that the sparsity degree varies with both model sizes and architectures.

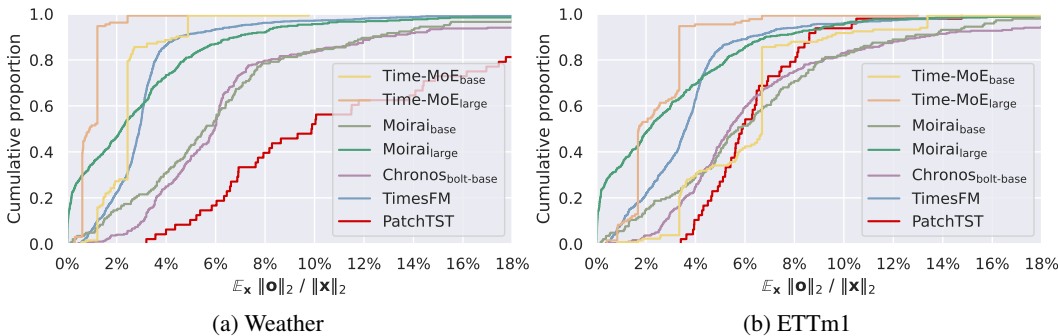

(a) Weather

(b) ETTm1

Fig. 2: Cumulative distribution of the average relative output norm of one attention head over the Weather and ETTm1 datasets.

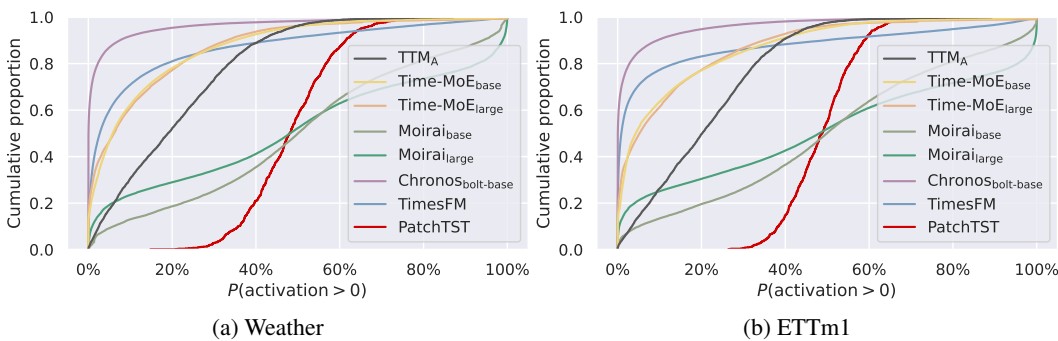

(a) Weather

(b) ETTm1

Fig. 3: Cumulative distribution of the activation probability of one FFN intermediate channel over the Weather and ETTm1 datasets.

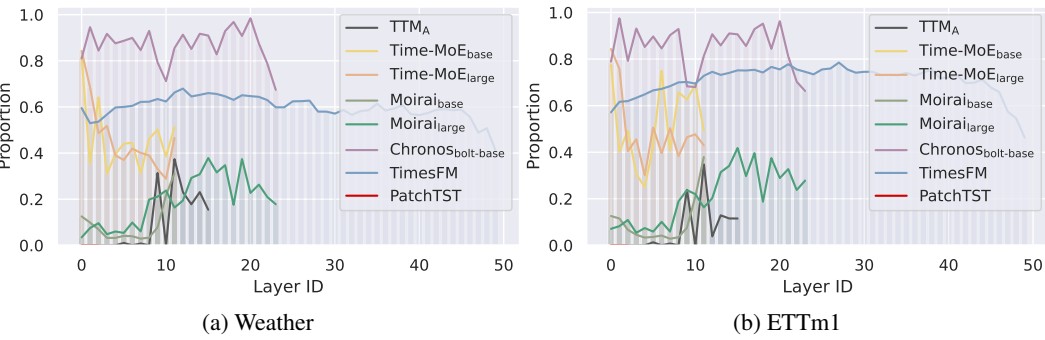

(a) Weather

(b) ETTm1

Fig. 4: Proportion of sparsely activated channels in each FFN layer. A channel is identified as sparsely activated if its activation probability is less than 5% over the downstream dataset.

### 3.2.2 Sparse Activations in FFNs

Shi et al. and Liu et al. found that MoE-based TSFMs tend to allocate different expert FFNs for samples from diverse datasets, and sparsely activated experts can address the heterogeneity of time series. We assume that there is also a sparse activation pattern at the granularity of FFN channels.

**Definition 2** The *activation probability* of the $i$-th intermediate channel of FFNs is defined as the frequency of its activations (denoted by $a_i$) being positive, i.e., $P(a_i > 0)$, over the dataset.

Fig. 3 highlights distinct FFN activation patterns. PatchTST utilizes their FFN channels relatively densely, with all channels active at least 20% computations. By contrast, several TSFMs exhibit extreme sparsity: 20-60% of FFN channels in Chronos and Time-MoE models remain entirely inactive on the downstream data. Notably, even non-MoE architectures like Chronos and Moirai demonstrate this ability to selectively activate FFN parameter subgroups for specific tasks. This

sparsity tends to increase with model size, as seen with Moirai$_{\text{large}}$ being sparser than Moirai$_{\text{base}}$. Even the smaller TTM (5M parameters) shows some FFN channels with low activation probabilities. Defining channels with <5% activation probability as unimportant, Fig. 4 further illustrates that significant FFN sparsity patterns are prevalent across different layers within TSFMs.

According to our analysis on attention heads and FFN activations, most pre-trained TSFMs exhibit inherent sparsity during inference, with a substantial portion of parameters being either redundant or inactive. Moreover, TSFMs exhibit different sparsity patterns over the Weather and ETTm1 datasets. We hypothesize that **their impressive zero-shot forecasting performance stems from the ability to selectively activate a task-relevant subset of parameters, leveraging prior knowledge.** This encourages us to explore more fine-grained sparsity estimations and universal pruning methods.

## 4 Methodology

In this section, we first define the pruning unit and its importance score, followed by a progressive pruning and fine-tuning method to specialize TSFMs for downstream tasks.

### 4.1 Pruning Unit

For a comprehensive removal of task-irrelevant parameters, we define a pruning unit (i.e., the minimal pruning elements) as the input/output channel of any linear layer, covering self-attention modules and FFNs. Formally, a linear layer $f(\cdot)$ is parameterized by a transformation matrix $\mathbf{W} \in \mathbb{R}^{d_{\text{in}} \times d_{\text{out}}}$ and (optionally) a bias term $\mathbf{b} \in \mathbb{R}^{d_{\text{out}}}$, where $d_{\text{in}}$ and $d_{\text{out}}$ are the dimension of the input and output channels, respectively. Additionally, we apply two binary mask vectors $\mathbf{m}^{\text{in}} \in \{0, 1\}^{d_{\text{in}}}$ and $\mathbf{m}^{\text{out}} \in \{0, 1\}^{d_{\text{out}}}$ to the input and output of $f(\cdot)$, respectively, which are initialized as $\mathbf{1}$. Given an input $\mathbf{x} \in \mathbb{R}^{\text{in}}$, the masked output $\mathbf{h} \in \mathbb{R}^{\text{out}}$ is defined as below:

$$\mathbf{h} = f(\mathbf{x} \odot \mathbf{m}^{\text{in}}) \odot \mathbf{m}^{\text{out}} = \mathbf{x} \left( \mathbf{W} \odot (\mathbf{m}^{\text{in}\top} \mathbf{m}^{\text{out}}) \right) + \mathbf{b} \odot \mathbf{m}^{\text{out}}, \tag{4}$$

where $\odot$ corresponds to the element-wise product. We can prune the $i$-th row of $\mathbf{W}$ by setting $m_i^{\text{in}}$, the $i$-th element of $\mathbf{m}^{\text{in}}$, to zero. Likewise, the $j$-th column of $\mathbf{W}$ and the $j$-th element of $\mathbf{b}$ will be pruned if $m_j^{\text{out}} = 0$. It is worth mentioning that due to the residual connection and multi-head attention in Transformers, masking the output channels of a given layer is not equivalent to masking the input channels in the subsequent layer. Therefore, we treat the masking operations on input and output channels independently.

### 4.2 Importance Score of Pruning Unit

From the perspective of forecasting loss, a redundant channel, if pruned, would result in negligible loss changes. Following prior works [22], we define the channel-wise importance score as follows:

**Definition 3** The *importance score* of channel $i$, denoted as $s_i$, is defined by $s_i = |\mathcal{L}(\mathbf{m} - \boldsymbol{e}_i) - \mathcal{L}(\mathbf{m})|$, where $\boldsymbol{e}_i$ denote a one-hot vector with the $i$-th element equal to 1.

For computational efficiency, we derive an approximation of $s_i$ by:

$$s_i \approx \left| -\boldsymbol{e}_i^\top \nabla_{\mathbf{m}} \mathcal{L} + \frac{1}{2} \boldsymbol{e}_i^\top \left( \nabla_{\mathbf{m}}^2 \mathcal{L} \right) \boldsymbol{e}_i \right| = \left| -\frac{1}{N} \sum_{n=1}^N \frac{\partial \mathcal{L}_n}{\partial m_i} + \frac{1}{2N} \sum_{n=1}^N \frac{\partial^2 \mathcal{L}_n}{\partial m_i^2} \right| \tag{5}$$

$$\approx \left| -\frac{1}{N} \sum_{n=1}^N \frac{\partial \mathcal{L}_n}{\partial m_i} + \frac{1}{2N} \sum_{n=1}^N \left( \frac{\partial \mathcal{L}_n}{\partial m_i} \right)^2 \right|, \tag{6}$$

where $N$ is the number of samples involved in calculating forecast loss $\mathcal{L}$, and $\mathcal{L}_n$ is the forecasting loss of the $n$-th sample. In Eq. (5), we apply the second-order Taylor expansion to the forecasting loss $\mathcal{L}$ to efficiently evaluate the loss change. In Eq. (6), we employ Fisher information to approximate the second-order derivative by the square of the first-order derivative [18].

The derivative w.r.t. $m_i^{\text{in}}$ is computed by $\partial \mathcal{L} / \partial m_i^{\text{in}} = x_i \cdot \partial \mathcal{L} / \partial (x_i m_i^{\text{in}})$. In a special case when $x_i$ is always a zero activation, the $i$-th input channel has a zero importance score and will be pruned. The derivative of $\mathbf{m}^{\text{out}}$ can be computed by $\partial \mathcal{L} / \partial \mathbf{m}^{\text{out}} = f(\mathbf{x} \odot \mathbf{m}^{\text{in}}) \cdot \partial \mathcal{L} / \partial \mathbf{h}$. The output channels

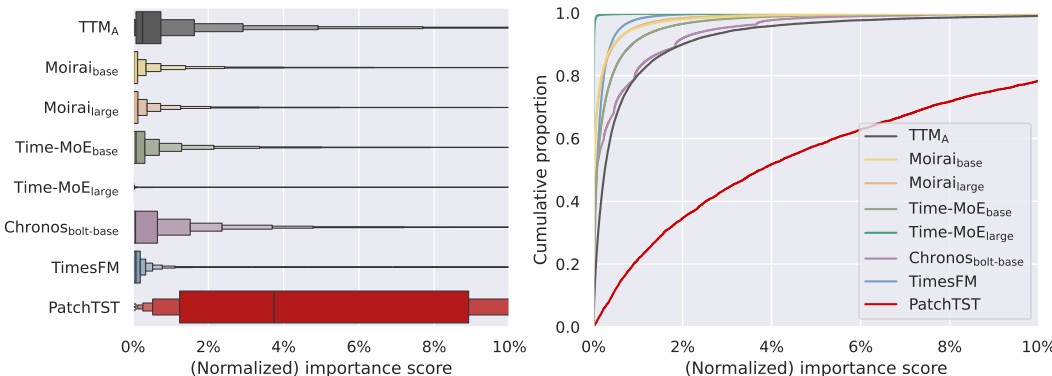

Fig. 5: *(Left)* Boxplot of importance scores, where the largest box encompasses the middle 50% of the data distribution. *(Right)* Cumulative distribution of importance scores. The scores are calculated based on the Weather dataset and further divided by the maximum value. Distributions on the ETTm1 dataset are provided in Fig. 8.

of an attention head will also have low importance scores if the outputs are marginal and the loss derivatives are small. On top of the inherent sparsity in TSFMs, the importance score metric can also measure other redundant channels whose outputs are of great value but get counteracted by other channels or activation functions in deeper layers. Therefore, we adopt the importance score for a comprehensive pruning to extract the critical network structure.

**Analysis.** Fig. 5 illustrates the distribution of channel importance scores over the Weather dataset. The long-tailed distribution of these scores within TSFMs demonstrates significant redundancy in channels. This channel redundancy is notably more pronounced in larger TSFM variants. This TSFM behavior caters to the nature of temporal data, where the presence of concept drift [6, 10, 14, 32–34] implies that enduringly effective data patterns are scarce. Consequently, structured pruning emerges as a compelling strategy to selectively restrain the number of exploited latent features.

## 4.3 Progressive Pruning and Then Finetune

To avoid over-pruning at one time, it is desirable to remove a small number of channels at one pruning step and re-calculate the loss at the next step. Given the high expense of inferring the whole training set repeatedly, we adopt pruning based on batches. First, sampling the $j$-th batch, we estimate the current importance score $s^{(j)}$ by Eq. (6). Second, given previously calculated scores, we estimate an exponential moving average (EMA) by $\tilde{s}^{(j)} = \alpha s^{(j)} + (1 - \alpha)\tilde{s}^{(j-1)}$, where $\alpha$ is a hyperparameter. Third, globally across all model layers, we rank $\tilde{s}^{(j)}$ in ascending order and remove unpruned channels with the lowest importance based on a predefined pruning ratio. Finally, we obtain a pruned model after one or a few epochs over the training set. Optionally for acceleration, one can first prune out insignificant attention heads and sparse FFN channels discovered in the forward pass based on a predefined threshold. Then, we involve gradients to compute importance scores of other channels for a fine-grained pruning. After pruning, we fine-tune the remaining parameters on the downstream data.

---

**Algorithm 1** Prune-then-Finetune
---
**Require:** Forecaster $\mathcal{F}$ with $C$ channels, data $\mathcal{D}$, pruning number per batch $K$, EMA coefficient $\alpha$
    **for** $i = 1$ to $C$ **do**
        Initialize mask $m_i \leftarrow 1$ and score $s_i^0 \leftarrow 0$;
    **end for**
    **for** $j = 1$ to $B$ **do**
        Sample batch $\mathcal{D}_j$ from $\mathcal{D}$;
        Forecast $\mathcal{D}_j$ with masks $\{m_i\}_i^C$;
        Calculate loss gradients w.r.t. each $m_i$;
        **for** $i = 1$ to $C$ **do**
            Calculate scores $s_i^{(j)}$ using Eq. (6);
            $\tilde{s}_i^{(j)} \leftarrow \alpha s_i^{(j)} + (1 - \alpha)\tilde{s}_i^{(j-1)}$;
        **end for**
        **for** $i = 1$ to $C$ **do**
            **if** $\tilde{s}_i^{(j)} \in TopK(\{-\tilde{s}_k^{(j)} \mid m_k = 1\})$ **then**
                $m_i \leftarrow 0$;
            **end if**
        **end for**
    **end for**
Prune channels of which $m_i = 0$;
Fine-tune the unpruned parameters of $\mathcal{F}$ on $\mathcal{D}$.

---

# 5 Experiments

In this section, we present extensive experiments to evaluate the effectiveness of structured pruning when adapting TSFMs to downstream tasks. We compare full-shot forecasting performance with and without pruning, analyze the resulting sparsity patterns and inference speedups, and assess the transferability of the pruned TSFMs to other datasets. Additional experimental results are provided in Appendix C, including short-term forecasting results on the M4 benchmark [23] and ablation studies on the tuning methods (Table 8) and the pruning methods (Table 9).

## 5.1 Experimental Settings

**Foundation Models.** Our experiments are conducted on a range of pre-trained Transformer-based time series forecasting models (TSFMs), including Time-MoE$_{base}$[1], Time-MoE$_{large}$[2], Timer-XL[3], Moirai$_{base}$[4], Moirai$_{large}$[5], Chronos-bolt$_{base}$[6], and TTM$_A$[7]. We inherit the open-source codes and pre-trained checkpoints from their official repositories.

**Datasets.** We evaluate long-term forecasting performance on three widely used benchmarks: ETT [35], Weather [31], and Electricity [15]. Dataset statistics and train/validation/test splits are summarized in Table 5. Pruning and fine-tuning are performed on the training sets, and we report forecasting accuracy on the test sets using Mean Squared Error (MSE) and Mean Absolute Error (MAE) over horizons of 96, 192, 336, and 720 steps.

**Implementation Details.** To ensure fair comparison, we carefully tune all models' hyperparameters (e.g., learning rate, pruning ratio) on the validation set. For autoregressive TSFMs, a single model is trained and pruned per dataset and reused across all forecasting lengths. In contrast, for non-autoregressive models (e.g., Moirai and TTM), we train separate models optimized for each forecasting horizon. Further details on the hyperparameter search process are provided in Appendix A.2 to facilitate reproducibility. For simplicity, our main pruning strategy relies solely on importance scores. Additional results using alternative pruning criteria are presented in the supplementary material.

## 5.2 Forecasting Performance

As shown in Table 2, our prune-then-finetune paradigm achieves higher predictive performance than directly fine-tuning TSFMs in 83% of forecasting tasks. The prediction errors are reduced by an average of 4.4%, 4.0%, 2.5%, 3.2%, 1.3%, 3.3%, 1.0% for Time-MoE$_{base}$, Time-MoE$_{large}$, Timer-XL, Moirai$_{base}$, Moirai$_{large}$, Chronos$_{bolt-base}$, TTM$_A$, respectively. The maximum reduction is 22.8% (for Time-MoE$_{base}$ on ETTm2), while the maximum performance drop is capped at only 1%. Notably, our method significantly increases the win rate of several TSFMs against PatchTST, with the strong baseline surpassed on all benchmarks, showcasing the remarkable potential of structured pruning. As for autoregressive predictions of TimeMoE, Timer-XL, and Chronos, their performance gains from pruning become more pronounced with longer forecasting horizons, due to reduced error accumulation. We posit that our pruning method goes beyond the inherent sparsity of TSFMs by deactivating more channels to regularize exploiting latent features, while fine-tuning the original model may mistakenly focus on outdated or noisy features. Different from random dropout, pruning focuses on a task-relevant subspace of parameters consistently across fine-tuning and inference.

## 5.3 Sparsification Analysis

As shown in Fig. 6, we compare the sparsification results in terms of the pruning ratios across input and output channels. Although both models consist of the same number of Transformer layers, Moirai$_{large}$ adopts an encoder-only architecture, while Chronos$_{bolt-base}$ follows an encoder-decoder

---

[1] https://huggingface.co/Maple728/TimeMoE-50M

[2] https://huggingface.co/Maple728/TimeMoE-200M

[3] https://huggingface.co/thuml/timer-base-84m

[4] https://huggingface.co/Salesforce/Moirai-1.0-R-base

[5] https://huggingface.co/Salesforce/Moirai-1.0-R-large

[6] https://huggingface.co/amazon/chronos-bolt-base

[7] https://huggingface.co/ibm-research/ttm-research-r2

Table 2: Full results of forecasting performance. "FT" is short for fine-tuning. Results of PatchTST are obtained from Li et al.. A lower MSE or MAE indicates a better prediction. For each comparison between results without and with pruning, we mark the winner in **bold**. For each prediction task, we mark the best results in yellow. Zero-shot performance is shown in Table 10.

| Model | TimeMoE$_{base}$ | | | | TimeMoE$_{large}$ | | | | Timer-XL | | | | Moirai$_{base}$ | | | | Moirai$_{large}$ | | | | Chronos$_{bolt\text{-}base}$ | | | | TTM$_A$ | | | | PatchTST | |
|---|---|---|---|---|---|---|---|---|---|---|---|---|---|---|---|---|---|---|---|---|---|---|---|---|---|---|---|---|---|---|
| Method | FT | | Prune+FT | | FT | | Prune+FT | | FT | | Prune+FT | | FT | | Prune+FT | | FT | | Prune+FT | | FT | | Prune+FT | | FT | | Prune+FT | | | |
| Tasks | MSE | MAE | MSE | MAE | MSE | MAE | MSE | MAE | MSE | MAE | MSE | MAE | MSE | MAE | MSE | MAE | MSE | MAE | MSE | MAE | MSE | MAE | MSE | MAE | MSE | MAE | MSE | MAE | MSE | MAE |
| **ETTh1** | | | | | | | | | | | | | | | | | | | | | | | | | | | | | | |
| 96 | 0.340 | 0.377 | 0.340 | **0.376** | 0.331 | 0.372 | **0.329** | **0.372** | 0.365 | 0.393 | **0.353** | **0.391** | 0.367 | 0.396 | **0.366** | 0.396 | 0.371 | 0.391 | **0.370** | 0.391 | 0.366 | **0.374** | 0.366 | 0.375 | 0.359 | 0.395 | **0.358** | **0.393** | 0.376 | 0.396 |
| 192 | 0.380 | 0.408 | **0.376** | **0.404** | 0.371 | 0.398 | **0.369** | 0.399 | 0.400 | 0.416 | **0.389** | **0.413** | 0.405 | 0.418 | **0.404** | 0.418 | 0.416 | 0.416 | **0.413** | 0.416 | 0.418 | 0.405 | **0.417** | 0.405 | 0.392 | 0.417 | **0.391** | **0.410** | 0.399 | 0.416 |
| 336 | 0.406 | 0.432 | **0.399** | **0.425** | 0.392 | 0.413 | **0.389** | **0.411** | 0.419 | 0.431 | **0.409** | **0.426** | 0.435 | 0.434 | **0.429** | **0.430** | 0.455 | 0.431 | 0.455 | 0.432 | 0.453 | 0.424 | **0.449** | **0.422** | 0.407 | 0.430 | **0.402** | **0.424** | 0.418 | 0.432 |
| 720 | 0.437 | 0.469 | **0.424** | **0.457** | 0.403 | 0.438 | **0.387** | **0.419** | 0.454 | 0.465 | **0.433** | **0.455** | 0.465 | 0.459 | **0.427** | **0.441** | 0.475 | 0.457 | **0.454** | **0.444** | 0.452 | 0.441 | **0.451** | **0.437** | 0.448 | 0.467 | 0.449 | 0.465 | 0.450 | 0.469 |
| Avg | 0.391 | 0.422 | **0.385** | **0.416** | 0.374 | 0.405 | **0.369** | **0.400** | 0.410 | 0.426 | **0.396** | **0.421** | 0.418 | 0.427 | **0.407** | **0.421** | 0.429 | 0.424 | **0.423** | **0.421** | 0.422 | 0.411 | **0.421** | **0.410** | 0.402 | 0.427 | **0.400** | **0.423** | 0.411 | 0.428 |
| #p | | | 63% | | | | 63% | | | | 67% | | | | 58%~69% | | | | 36%~39% | | | | 55% | | | | 91%~93% | | | |
| **ETTh2** | | | | | | | | | | | | | | | | | | | | | | | | | | | | | | |
| 96 | 0.267 | 0.330 | **0.262** | **0.327** | 0.272 | 0.332 | 0.272 | 0.332 | 0.287 | 0.345 | **0.275** | **0.337** | 0.273 | 0.338 | **0.272** | 0.338 | 0.288 | 0.341 | **0.287** | 0.341 | 0.266 | 0.311 | 0.266 | 0.311 | 0.264 | 0.333 | **0.262** | **0.328** | 0.277 | 0.339 |
| 192 | 0.341 | 0.380 | **0.331** | **0.374** | 0.356 | 0.383 | **0.356** | **0.382** | 0.344 | 0.385 | **0.334** | **0.378** | 0.331 | 0.385 | 0.331 | 0.385 | 0.349 | 0.386 | **0.350** | 0.387 | 0.334 | 0.357 | **0.331** | **0.356** | 0.321 | 0.372 | 0.324 | **0.370** | 0.345 | 0.381 |
| 336 | 0.386 | 0.415 | **0.372** | **0.403** | 0.420 | 0.424 | **0.411** | **0.418** | 0.375 | 0.410 | **0.368** | **0.404** | 0.358 | 0.410 | **0.358** | **0.409** | 0.373 | 0.409 | **0.371** | **0.408** | 0.373 | 0.410 | **0.370** | **0.388** | 0.351 | 0.397 | **0.351** | **0.396** | 0.368 | 0.404 |
| 720 | 0.419 | 0.454 | **0.377** | **0.423** | 0.534 | 0.494 | **0.459** | **0.454** | 0.411 | 0.445 | **0.398** | **0.439** | 0.392 | 0.444 | **0.388** | **0.442** | 0.402 | 0.439 | **0.394** | **0.435** | 0.395 | 0.418 | **0.389** | **0.415** | 0.395 | 0.439 | **0.393** | **0.437** | 0.397 | 0.432 |
| Avg | 0.353 | 0.395 | **0.336** | **0.382** | 0.396 | 0.408 | **0.375** | **0.397** | 0.354 | 0.396 | **0.344** | **0.390** | 0.339 | 0.394 | **0.337** | **0.394** | 0.353 | 0.394 | **0.351** | **0.393** | 0.343 | 0.369 | **0.339** | **0.368** | 0.333 | 0.385 | **0.333** | **0.383** | 0.347 | 0.389 |
| #p | | | 82% | | | | 77% | | | | 68% | | | | 55%~78% | | | | 27%~36% | | | | 56% | | | | 93%~98% | | | |
| **ETTm1** | | | | | | | | | | | | | | | | | | | | | | | | | | | | | | |
| 96 | 0.220 | 0.302 | 0.220 | 0.302 | 0.214 | 0.305 | **0.211** | **0.299** | 0.305 | 0.355 | **0.297** | **0.354** | 0.309 | 0.343 | **0.304** | **0.342** | 0.311 | 0.345 | 0.312 | 0.346 | 0.285 | 0.315 | **0.282** | 0.315 | 0.295 | 0.346 | **0.291** | **0.340** | 0.290 | 0.343 |
| 192 | 0.284 | 0.347 | **0.282** | **0.344** | 0.268 | 0.345 | **0.257** | **0.333** | 0.348 | 0.383 | **0.337** | **0.381** | 0.344 | 0.370 | **0.342** | 0.370 | 0.343 | 0.370 | **0.342** | 0.370 | 0.329 | 0.345 | **0.323** | **0.343** | 0.332 | 0.367 | **0.327** | **0.363** | 0.329 | 0.368 |
| 336 | 0.349 | 0.392 | **0.335** | **0.381** | 0.326 | 0.386 | **0.300** | **0.364** | 0.382 | 0.406 | **0.367** | **0.403** | 0.379 | 0.385 | **0.375** | **0.384** | 0.364 | 0.385 | **0.359** | 0.385 | 0.361 | 0.369 | **0.350** | **0.365** | 0.356 | 0.383 | 0.356 | **0.381** | 0.360 | 0.390 |
| 720 | 0.462 | 0.466 | **0.415** | **0.437** | 0.428 | 0.449 | **0.380** | **0.415** | 0.428 | 0.439 | **0.409** | **0.434** | 0.410 | 0.409 | **0.401** | 0.409 | 0.400 | 0.409 | 0.400 | 0.411 | 0.433 | 0.415 | **0.412** | **0.404** | 0.409 | 0.420 | **0.395** | **0.406** | 0.416 | 0.422 |
| Avg | 0.329 | 0.377 | **0.313** | **0.366** | 0.309 | 0.371 | **0.287** | **0.353** | 0.366 | 0.396 | **0.353** | **0.393** | 0.362 | 0.376 | **0.354** | **0.375** | 0.355 | 0.377 | **0.353** | **0.378** | 0.352 | 0.361 | **0.342** | **0.357** | 0.348 | 0.379 | **0.342** | **0.373** | 0.349 | 0.381 |
| #p | | | 92% | | | | 64% | | | | 81% | | | | 51%~60% | | | | 51%~52% | | | | 18% | | | | 86%~99% | | | |
| **ETTm2** | | | | | | | | | | | | | | | | | | | | | | | | | | | | | | |
| 96 | 0.161 | 0.253 | **0.158** | **0.251** | 0.166 | 0.258 | **0.164** | 0.258 | 0.191 | 0.280 | **0.181** | **0.273** | 0.174 | 0.261 | **0.172** | **0.259** | 0.165 | 0.251 | 0.165 | 0.251 | 0.160 | 0.231 | **0.156** | 0.234 | 0.160 | 0.251 | 0.160 | **0.248** | 0.165 | 0.254 |
| 192 | 0.229 | 0.308 | **0.217** | **0.299** | 0.248 | 0.318 | **0.236** | **0.313** | 0.245 | 0.320 | **0.233** | **0.310** | 0.234 | 0.304 | **0.231** | **0.300** | 0.234 | 0.304 | **0.232** | **0.301** | 0.225 | 0.276 | **0.215** | **0.275** | 0.213 | 0.289 | 0.213 | **0.286** | 0.221 | 0.292 |
| 336 | 0.313 | 0.367 | **0.273** | **0.342** | 0.347 | 0.381 | **0.314** | **0.367** | 0.297 | 0.356 | **0.283** | **0.344** | 0.284 | 0.337 | 0.287 | 0.340 | 0.286 | 0.317 | **0.268** | **0.313** | 0.286 | 0.316 | **0.269** | 0.322 | 0.269 | 0.326 | **0.268** | 0.322 | 0.275 | 0.325 |
| 720 | 0.486 | 0.464 | **0.375** | **0.407** | 0.552 | 0.488 | **0.471** | **0.453** | 0.389 | 0.416 | **0.367** | **0.396** | 0.353 | 0.386 | **0.350** | **0.384** | 0.366 | 0.397 | **0.365** | **0.396** | 0.381 | 0.377 | **0.348** | **0.367** | 0.346 | 0.379 | **0.338** | **0.369** | 0.360 | 0.380 |
| Avg | 0.297 | 0.348 | **0.256** | **0.325** | 0.328 | 0.361 | **0.296** | **0.348** | 0.281 | 0.343 | **0.266** | **0.331** | 0.261 | 0.323 | **0.259** | **0.320** | 0.263 | 0.324 | **0.262** | **0.322** | 0.263 | 0.300 | **0.247** | **0.297** | 0.247 | 0.311 | **0.245** | **0.306** | 0.255 | 0.313 |
| #p | | | 75% | | | | 57% | | | | 64% | | | | 51%~56% | | | | 35%~38% | | | | 18% | | | | 86%~99% | | | |
| **Weather** | | | | | | | | | | | | | | | | | | | | | | | | | | | | | | |
| 96 | 0.141 | **0.194** | 0.141 | 0.196 | 0.140 | 0.193 | **0.139** | **0.190** | 0.169 | 0.226 | **0.165** | **0.219** | 0.178 | 0.228 | **0.159** | **0.222** | 0.147 | 0.199 | **0.147** | **0.198** | 0.174 | 0.209 | **0.143** | **0.180** | 0.150 | 0.202 | **0.145** | **0.195** | 0.149 | 0.196 |
| 192 | 0.196 | 0.254 | **0.191** | **0.250** | 0.191 | 0.245 | **0.189** | **0.241** | 0.217 | 0.271 | **0.209** | **0.261** | 0.220 | 0.269 | **0.216** | **0.265** | 0.211 | 0.263 | **0.198** | **0.255** | 0.231 | 0.258 | **0.186** | **0.223** | 0.192 | 0.243 | **0.188** | **0.240** | 0.193 | 0.240 |
| 336 | 0.260 | 0.309 | **0.249** | **0.301** | 0.255 | 0.297 | **0.253** | **0.292** | 0.267 | 0.308 | **0.255** | **0.296** | 0.308 | 0.330 | **0.238** | **0.287** | 0.277 | 0.315 | 0.279 | 0.317 | 0.289 | 0.298 | **0.235** | **0.262** | 0.237 | 0.279 | **0.236** | **0.277** | 0.244 | 0.281 |
| 720 | 0.346 | 0.378 | **0.331** | **0.370** | 0.371 | 0.383 | **0.361** | **0.370** | 0.336 | 0.358 | **0.322** | **0.345** | 0.390 | 0.363 | **0.333** | **0.361** | 0.373 | 0.383 | **0.345** | **0.365** | 0.378 | 0.347 | **0.296** | **0.306** | 0.314 | 0.335 | **0.313** | **0.335** | 0.314 | 0.332 |
| Avg | 0.236 | 0.284 | **0.228** | **0.279** | 0.239 | 0.280 | **0.226** | **0.273** | 0.247 | 0.291 | **0.237** | **0.280** | 0.274 | 0.304 | **0.237** | **0.284** | 0.252 | 0.290 | **0.242** | **0.284** | 0.268 | 0.278 | **0.215** | **0.243** | 0.223 | 0.265 | **0.221** | **0.262** | 0.225 | 0.262 |
| #p | | | 68% | | | | 88% | | | | 75% | | | | 20%~62% | | | | 20%~44% | | | | 17% | | | | 91%~94% | | | |
| **Electricity** | | | | | | | | | | | | | | | | | | | | | | | | | | | | | | |
| 96 | - | - | - | - | - | - | - | - | 0.139 | 0.237 | **0.138** | 0.237 | 0.152 | 0.247 | **0.135** | **0.236** | 0.148 | 0.243 | **0.138** | **0.236** | 0.121 | 0.208 | **0.120** | **0.206** | 0.129 | 0.226 | 0.129 | 0.226 | 0.133 | 0.233 |
| 192 | - | - | - | - | - | - | - | - | 0.158 | 0.256 | **0.157** | **0.255** | 0.170 | 0.263 | **0.162** | 0.263 | 0.172 | 0.265 | **0.164** | 0.265 | 0.139 | 0.226 | **0.138** | **0.224** | 0.147 | 0.243 | 0.148 | 0.243 | 0.150 | 0.248 |
| 336 | - | - | - | - | - | - | - | - | 0.177 | 0.275 | **0.176** | 0.275 | 0.192 | 0.285 | **0.176** | **0.275** | 0.157 | 0.244 | **0.155** | **0.242** | 0.157 | 0.243 | **0.156** | **0.241** | 0.167 | 0.266 | **0.165** | **0.263** | 0.168 | 0.267 |
| 720 | - | - | - | - | - | - | - | - | 0.219 | 0.313 | **0.217** | **0.312** | 0.220 | 0.307 | **0.210** | **0.305** | 0.219 | 0.309 | **0.215** | 0.310 | 0.199 | 0.283 | **0.194** | **0.277** | 0.202 | 0.297 | **0.199** | **0.293** | 0.202 | 0.295 |
| Avg | - | - | - | - | - | - | - | - | 0.173 | 0.270 | **0.172** | 0.270 | 0.184 | 0.276 | **0.171** | **0.270** | 0.182 | 0.274 | **0.173** | **0.274** | 0.154 | 0.240 | **0.152** | **0.237** | 0.161 | 0.258 | **0.160** | **0.256** | 0.163 | 0.261 |
| #p | | | - | | | | - | | | | 56% | | | | 40%~42% | | | | 3%~60% | | | | 53% | | | | 94%~98% | | | |
| **Win Rate** | 46.0% | | **68.0%** | | 46.0% | | **52.0%** | | 11.7% | | **26.7%** | | 25.0% | | **33.3%** | | 18.3% | | **20.0%** | | 80.0% | | **98.3%** | | 85.0% | | **93.3%** | | | |

* "#p" is the proportion of the remaining parameters after pruning relative to the original ones.
* "Win Rate" is based on comparisons between each TSFM and PatchTST over all datasets.
* Electricity included in the pre-training of Time-MoE is not evaluated, denoted by a dash (–).
* We present performance comparison about TimesFM$_{2.0\text{-}500M}$ and the Traffic [31] dataset in Table 9.

design. Both models generally exhibits more aggressive pruning in their deeper layers, with non-negligible pruning also observed in the shallow layers. In contrast, pruning in Chronos$_{bolt\text{-}base}$ is more concentrated in the decoder (i.e., the last 12 layers) than the encoder. We attribute this to its use of a much smaller patch size for time series tokenization, which requires the encoder to capture and aggregate more contextual information across tokens.

It is also important to note that channel pruning occurs across various layer types. Importance-score-based pruning successfully removes certain attention heads and sparsely activated FFN channels, as evidenced by the pruned input channels of $\mathbf{W}^O$ and $\mathbf{W}^{\text{down}}$, respectively. Furthermore, additional sparsification is introduced in other projection matrices, including $\mathbf{W}^Q$, $\mathbf{W}^K$, $\mathbf{W}^V$, and $\mathbf{W}^{\text{up}}$. The observed improvements in fine-tuning performance confirm the effectiveness of our method in systematically identifying and pruning redundant components using importance-based criteria.

## 5.4 Inference Efficiency

As shown in Table 3, in addition to the improvement in predictive performance, our proposed prune-then-finetune approach also gives a bonus in the speed of inference. We evaluate the computational efficiency of pruned TSFMs using the Weather benchmark. At each time step, all variates are processed as a batch, and we report the average CPU runtime during forecasting 720 steps, measured over 50 batches. Our results show that the pruned TSFMs achieve up to a 7.4× speedup, with the largest gain observed for Moirai$_{large}$.

Table 3: Inference time (in seconds) of TSFMs w/ and w/o pruning (horizon=720)

| | Original | Pruned | Speedup |
|---|---|---|---|
| TimeMoE$_{base}$ | 72 | **70** | 1.0× |
| TimeMoE$_{large}$ | 110 | **103** | 1.1× |
| Timer-XL | 0.18 | **0.16** | 1.1× |
| Moirai$_{base}$ | 0.37 | **0.14** | 2.6× |
| Moirai$_{large}$ | 1.86 | **0.25** | 7.4× |
| Chronos$_{bolt\text{-}base}$ | 8.51 | **1.26** | 6.5× |
| TTM$_A$ | 0.01 | 0.01 | 1.0× |

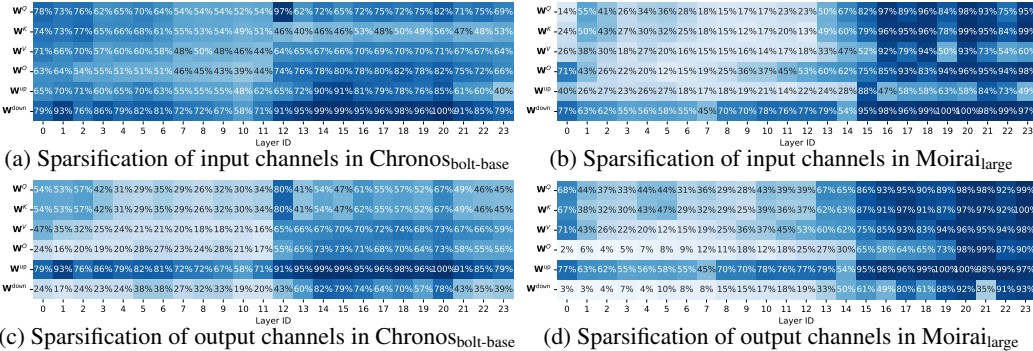

(a) Sparsification of input channels in Chronos$_{\text{bolt-base}}$     (b) Sparsification of input channels in Moirai$_{\text{large}}$

(c) Sparsification of output channels in Chronos$_{\text{bolt-base}}$     (d) Sparsification of output channels in Moirai$_{\text{large}}$

Fig. 6: Proportions of input/output channels pruned out in different layers of Moirai$_{\text{large}}$ and Chronos$_{\text{bolt-base}}$ on the Weather dataset. Results of other models are provided in Appendix C.2.

## 5.5 Transferability

We also observe that the pruned model exhibits strong transferability across related downstream tasks. We study Chronos using the ETTh and ETTm datasets from the electricity domain, with results summarized in Table 4. Notably, with appropriate source–target task pairs, the pruned model transferred from the source domain can even outperform the original (unpruned) model on the target task. This suggests that sparsity patterns learned in one domain may generalize well to semantically related tasks. Consequently, in data-scarce settings, one can leverage similar datasets to guide pruning and architecture specialization, potentially leading to improved performance.

Table 4: Transferability study (horizon=720)

| Model Methods | Chronos$_{\text{bolt-base}}$ | | | |
| | Zero-shot | | Prune | |
| Source→Target | MSE | MAE | MSE | MAE |
|---|---|---|---|---|
| ETTh1→ETTh2 | 0.420 | 0.421 | **0.389** | **0.410** |
| ETTh2→ETTh1 | 0.473 | 0.447 | **0.465** | **0.444** |
| ETTm1→ETTm2 | 0.420 | 0.395 | **0.396** | **0.385** |
| ETTm2→ETTm1 | 0.504 | 0.430 | **0.501** | **0.427** |
| ETTm1→ETTh2 | 0.420 | 0.421 | **0.408** | **0.418** |
| ETTm2→ETTh1 | 0.473 | 0.447 | **0.469** | **0.446** |
| ETTh1→ETTm2 | 0.420 | 0.395 | **0.376** | **0.376** |
| ETTh2→ETTm1 | 0.504 | 0.430 | **0.491** | **0.426** |

## 6 Conclusion

This paper sheds light on the critical challenge of adapting pre-trained TSFMs for superior performance on full-shot downstream forecasting tasks. Investigating inherent sparsity and redundancy within TSFMs, we suggest that the activated task-relevant subnetwork is valuable prior knowledge of a starting point for fine-tuning. We proposed a "prune-then-finetune" paradigm, employing structured pruning to identify and extract the critical subnetwork. Extensive experiments across seven TSFMs and six popular benchmarks compellingly demonstrated that our method significantly improves forecasting performance, enabling pruned TSFMs to surpass strong specialized baselines. This work pioneers structured pruning primarily for performance enhancement via architectural specialization, offering a promising pathway to unlock the full potential of TSFMs in practical applications.

## Acknowledgments and Disclosure of Funding

This work is supported by the National Key Research and Development Program of China (No. 2023YFB4502902), National Natural Science Foundation of China (No. 62522211), and Key Research and Development Program of Xinjiang Uygur Autonomous Region (Grant No. 2023B01027, 2023B01027-1).

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

# A   Implementation Details

## A.1   Benchmark Details

Table 5: Downstream forecasting dataset descriptions. Split denotes the number of time points in (train, validation, test) sets.

| Dataset | Variate | Dataset Split | Frequency | Variate Types |
|---|---|---|---|---|
| ETTh1 | 7 | (8545, 2881, 2881) | Hourly | electricity load, oil temperature |
| ETTh2 | 7 | (8545, 2881, 2881) | Hourly | electricity load, oil temperature |
| ETTm1 | 7 | (34465, 11521, 11521) | 15 min | electricity load, oil temperature |
| ETTm2 | 7 | (34465, 11521, 11521) | 15 min | electricity load, oil temperature |
| Weather | 21 | (36792, 5271, 10540) | 10 min | temperature, wind speed, pressure, rain, etc. |
| Electricity | 321 | (12280, 1755, 3509) | Hourly | electricity load |

We evaluate the long-term forecasting performance across six well-established datasets, including the Weather [31], Electricity [15], and ETT datasets (ETTh1, ETTh2, ETTm1, ETTm2) [35]. A detailed description of each dataset is provided in Table 5. We use the full test set without the "drop-last" strategy used in PatchTST [24]. We borrow benchmark results of PatchTST from TFB [26] that carefully tuned the hyperparameters.

## A.2   Experimental Details

We conduct extensive experiments based on 40 Nvidia V100 GPUs, 32 Nvidia 3090 GPUs, and Intel(R) Xeon(R) Platinum 8163 CPUs. The random seed is set to 1. The search grid of hyperparameters is provided in Table 6. Particularly, we set the fine-tuning epoch to 1 for Time-MoE, Chronos, and Moirai; otherwise, these models after multi-epoch fine-tuning usually achieve lower validation MSE but higher test MSE due to overfitting. To reduce tuning cost, we do not run all combinations of hyperparameters. For instance, we tune the hyperparameters of pruning and fine-tuning separately. As for pruning, we first tune the coefficient of exponential moving average that is optimal for a specific pruning ratio per epoch. Given four pruning ratios, we obtain four pruned models and tune the fine-tuning learning rates for each model. Alternatively, we can also select one pruned model that performs the best on the validation set and tune the learning rate of only a single model. The pruned models always follow the same batch size for fine-tuning the original model. The hyperparameter tuning for the most expensive model (e.g., Moirai$_{large}$) on the largest Electricity dataset consumes about 1 day.

Table 6: Grid of hyperparameters search

| | Time-MoE | TTM | Time-XL | Moirai | Chronos |
|---|---|---|---|---|---|
| Learning rate | \{1e-2, 1e-3, 1e-4, 1e-5, 1e-6, 1e-7, 1e-8, 1e-9\} | | | | |
| Context length | 4096 | 1536 | 2880 | \{2048, 3072, 4096\} | 2048 |
| Patch size | 1 | 1 | 96 | \{32, 64, 128\} | 16 |
| Batch size for fine-tuning | \{32, 64\} | | | \{256, 512\} | |
| Max. fine-tuning epoch | 1 | 10 | 10 | 1 | 1 |
| Patience for early stop | 1 | | | | |
| Pruning ratio per epoch | \{1%, 2%, 5%, 10%, 15%, 20%\} | | | | |
| Exponential moving average | \{0.1, 0.2, 0.4, 0.5, 0.6, 0.8\} | | | | |
| Batch size for pruning | 8192 | | | | |
| Max. pruning epoch | 10 | 1 | 10 | 10 | 1 |

# B    Additional Empirical Study

## B.1    Insignificant Weight Magnitudes

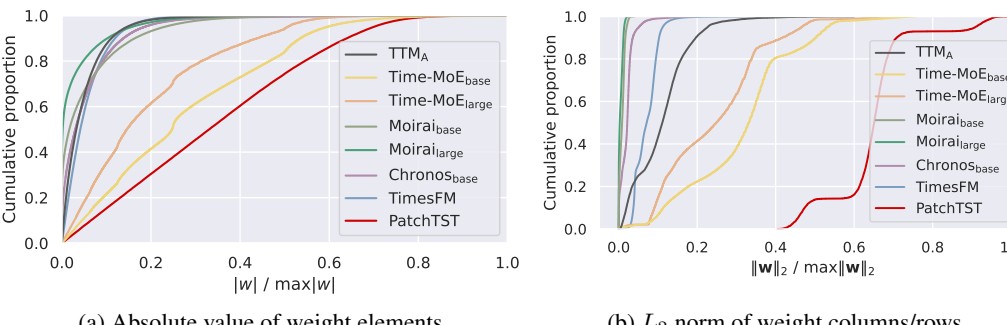

(a) Absolute value of weight elements            (b) $L_2$ norm of weight columns/rows

Fig. 7: Cumulative distribution of weight magnitudes, which are divided by the maximum value for a comparable study across models.

On top of the analysis of sparse attention heads and FFN channels, another question is whether all parameters of TSFMs are of vital importance. To ascertain it, we analyze the distribution of parameter magnitudes, which are defined as the absolute value of each element in weight matrices. For a comparable study across models, we normalize the magnitude by dividing each by the maximum value, ensuring all values fall within the [0, 1] range. As shown in Fig. 7a, Chronos and Moirai, based on Transformer encoders, have a considerable proportion of tiny parameter weights. In contrast, PatchTST has a nearly uniform distribution of weight magnitudes, though it is encoder-based as well. As for decoder-only TSFMs, Timer-XL and Time-MoE also have a skewed distribution of weight magnitudes. As for the vector-wise magnitude of parameter weights (i.e., the rows or columns of weight matrices), we depict the cumulative distribution in Fig. 7b, where the difference between TSFMs and PatchTST becomes more significant. These findings suggest that zero-shot forecasting may be based on a portion of parameters in TSFMs instead of all parameters. Nevertheless, it is worth mentioning that pruning based on magnitude is task-agnostic and may result in suboptimal downstream performance. This is why we adopt loss change to compute importance scores.

## B.2    Importance Scores over the ETTm1 dataset

We depict the distribution of importance scores over the ETTm1 dataset in Fig. 8. Compared with results on Weather as shown in Fig. 5, the redundancy patterns of the pre-trained TSFMs vary on the datasets, indicating that there are some task-specific parameters.

# C    Additional Experimental Results

## C.1    Short-term Forecasting Performance

Table 7 compares short-term forecasting performance of Chronos with different tuning methods over the M4 benchmark [23]. Better performance can be indicated by smaller evaluation metrics including MSE (mean-square error), MAE (mean absolute error), MASE (mean absolute scaled error), CRPS (continuous ranked probability score), and NRMSE (normalized root-mean-square error), where MSE and MAE are calculated based on raw values rather than normalized ones. In most cases, our proposed method can lead to better short-term forecasting performance than the naive full fine-tuning method.

## C.2    Sparsification Visualization

We visualize more sparsification results in Fig. 9. Moirai$_{base}$ and Timer-XL exhibit sparsification in input channels of output projections and intermediate channels of FFNs, which confirms that pruning based solely on importance scores can handle the inherent sparsity of TSFMs. In contrast, the sparsification of Time-MoE is distinct by mainly pruning the FFNs. This can be attributed to the fact

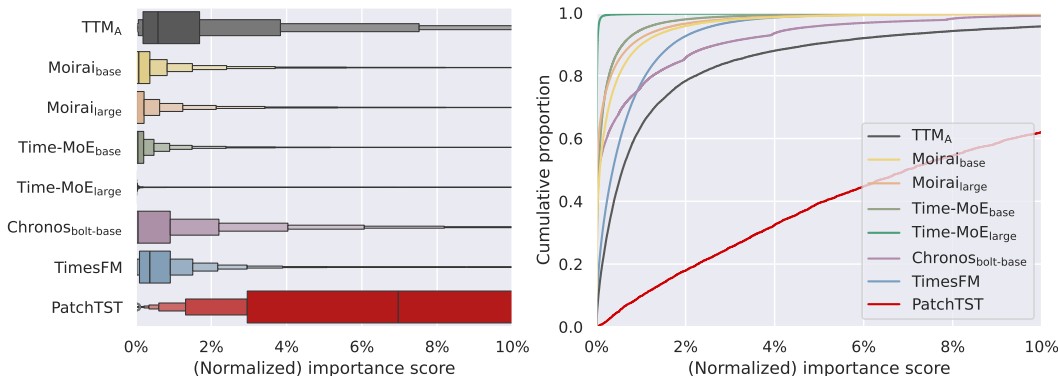

Fig. 8: (*Left*) Boxplot of importance scores, where the largest box encompasses the middle 50% of the data distribution. (*Right*) Cumulative distribution of importance scores. The scores are calculated based on the ETTm1 dataset and further divided by the maximum value.

Table 7: Short-term Forecasting Performance over the M4 benchmark.

| | Chronos | MSE | MAE | MASE | CRPS | NRMSE |
|---|---|---|---|---|---|---|
| **Yearly** (*H*=6) | Zero-shot | 3926423 | 943 | 3.507 | 0.121 | 0.318 |
| | Fine-Tune | 3226941 | 833 | 3.089 | 0.107 | 0.309 |
| | Prune+FT | **3125398** | **816** | **3.007** | **0.104** | **0.283** |
| **Quarterly** (*H*=8) | Zero-shot | 1878597 | 578 | 1.224 | 0.077 | 0.229 |
| | Fine-Tune | 1775734 | 571 | **1.221** | 0.075 | 0.223 |
| | Prune+FT | **1749123** | **551** | 1.225 | **0.074** | **0.221** |
| **Monthly** (*H*=18) | Zero-shot | 1934416 | 565 | 0.949 | 0.094 | 0.289 |
| | Fine-Tune | 1894232 | 571 | 0.946 | 0.095 | 0.286 |
| | Prune+FT | **1820581** | **556** | **0.920** | **0.091** | **0.280** |
| **Weekly** (*H*=13) | Zero-shot | 240658 | 258 | 2.08 | 0.038 | 0.089 |
| | Fine-Tune | 239732 | 257 | 2.07 | 0.038 | 0.089 |
| | Prune+FT | **238142** | **253** | **1.99** | **0.037** | 0.089 |
| **Daily** (*H*=14) | Zero-shot | **359007** | 172 | 3.20 | **0.093** | 0.021 |
| | Fine-Tune | 511617 | 175 | 3.21 | 0.110 | 0.021 |
| | Prune+FT | 536501 | **168** | **3.05** | 0.113 | 0.021 |
| **Hourly** (*H*=48) | Zero-shot | 1154587 | 244 | 0.837 | 0.025 | 0.147 |
| | Fine-Tune | 1032867 | 239 | 0.943 | 0.026 | 0.139 |
| | Prune+FT | **937494** | **227** | **0.805** | **0.024** | **0.131** |

that the usage of expert FFNs in MoE models is less frequent than the multi-head attention modules, resulting in less contribution to the forecast loss over a batch. Moreover, the multiple experts may share redundancy with each other, and pruning an expert may not lead to significant loss changes. In our supplementary materials, we will provide results of pruning the attention heads based on a threshold of the relative output norm.

### C.3 Performance of Other Tuning Methods

Parameter-Efficient Fine-Tuning (PEFT) has become a popular approach to efficient LLM fine-tuning by reducing the number of trainable parameters. LoRA [13] is one of the most prevalent PEFT methods that learns two additional low-rank weight matrices as an adjustment to the original full-rank weight matrix at each layer. In this section, we apply LoRA to Chronos$_{\text{bolt-base}}$ and TTM$_A$ with the rank hyperparameter chosen from {32, 64, 128}. As shown in Table 8, we do not observe considerable performance improvement of LoRA compared to full fine-tuning in most cases.

It is noteworthy that our prune-then-finetune paradigm is orthogonal to any model tuning methods. We conduct additional experiments by incorporating LoRA into our paradigm. Concretely, the dimensions of LoRA matrices will depend on the remaining input/output dimensions of each linear

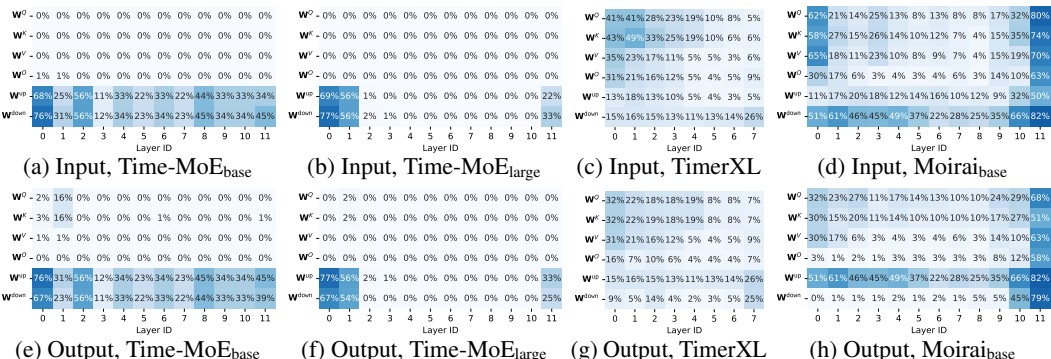

Fig. 9: Proportions of input/output channels pruned out in different layers of TSFMs.

Table 8: Performance comparisons between different tuning methods. $H$ denotes the forecast horizon. "FT" is short for fine-tuning.

| Models | | Chronos$_{base}$ | | | | | | | | TTM$_A$ | | | | | | | |
|---|---|---|---|---|---|---|---|---|---|---|---|---|---|---|---|---|---|
| Methods | | LoRA | | Full FT | | Prune+LoRA | | Prune+Full FT | | LoRA | | Full FT | | Prune+LoRA | | Prune+Full FT | |
| Datasets | H | MSE | MAE | MSE | MAE | MSE | MAE | MSE | MAE | MSE | MAE | MSE | MAE | MSE | MAE | MSE | MAE |
| ETTh1 | 96 | 0.366 | **0.374** | 0.366 | **0.374** | 0.366 | 0.375 | 0.366 | 0.375 | 0.360 | 0.396 | 0.359 | 0.395 | **0.358** | **0.393** | **0.358** | **0.393** |
| | 192 | 0.420 | 0.405 | 0.418 | 0.405 | **0.417** | 0.405 | **0.417** | 0.405 | 0.394 | 0.420 | 0.392 | 0.417 | **0.388** | **0.411** | 0.391 | 0.410 |
| | 336 | 0.456 | 0.425 | 0.453 | 0.424 | 0.450 | 0.422 | 0.449 | 0.422 | 0.409 | 0.433 | 0.407 | 0.430 | 0.406 | 0.424 | **0.402** | **0.424** |
| | 720 | 0.454 | 0.441 | 0.452 | 0.441 | 0.451 | 0.437 | 0.451 | 0.437 | 0.448 | 0.471 | 0.448 | 0.467 | **0.447** | **0.464** | 0.449 | 0.465 |
| | **Avg.** | 0.424 | 0.411 | 0.422 | 0.411 | **0.421** | 0.410 | **0.421** | 0.410 | 0.403 | 0.430 | 0.402 | 0.427 | **0.400** | 0.424 | **0.400** | 0.423 |
| ETTh2 | 96 | 0.267 | 0.312 | **0.266** | 0.311 | **0.266** | 0.311 | **0.266** | 0.311 | 0.264 | 0.333 | 0.264 | 0.333 | 0.262 | 0.329 | **0.262** | 0.328 |
| | 192 | 0.335 | 0.358 | 0.334 | 0.357 | 0.333 | 0.356 | 0.331 | 0.356 | **0.321** | 0.372 | **0.321** | 0.372 | 0.324 | 0.371 | 0.324 | 0.370 |
| | 336 | 0.379 | 0.392 | 0.378 | 0.391 | 0.376 | 0.389 | 0.370 | 0.388 | **0.351** | 0.397 | **0.351** | 0.397 | 0.352 | 0.396 | 0.351 | 0.396 |
| | 720 | 0.395 | 0.418 | 0.395 | 0.418 | 0.392 | **0.414** | 0.389 | 0.415 | 0.395 | 0.439 | 0.395 | 0.439 | **0.389** | **0.432** | 0.393 | 0.437 |
| | **Avg.** | 0.344 | 0.370 | 0.343 | 0.369 | 0.342 | 0.368 | **0.339** | 0.368 | 0.333 | 0.385 | 0.333 | 0.385 | **0.332** | **0.382** | 0.333 | 0.383 |
| ETTm1 | 96 | 0.285 | 0.315 | 0.285 | 0.315 | 0.285 | 0.315 | **0.283** | 0.315 | 0.295 | 0.346 | 0.295 | 0.346 | **0.289** | **0.339** | 0.291 | 0.340 |
| | 192 | 0.330 | **0.344** | 0.329 | 0.345 | 0.329 | **0.344** | 0.326 | **0.344** | 0.334 | 0.368 | 0.332 | 0.366 | **0.327** | 0.365 | **0.327** | 0.363 |
| | 336 | 0.363 | 0.368 | 0.361 | 0.369 | 0.362 | 0.367 | 0.356 | 0.367 | 0.358 | 0.384 | **0.356** | 0.383 | 0.357 | 0.382 | **0.356** | 0.381 |
| | 720 | 0.434 | 0.409 | 0.434 | 0.413 | 0.430 | **0.405** | 0.423 | 0.407 | **0.394** | 0.409 | 0.409 | 0.420 | 0.396 | **0.405** | 0.395 | 0.406 |
| | **Avg.** | 0.353 | 0.359 | 0.352 | 0.361 | 0.352 | **0.358** | 0.347 | **0.358** | 0.345 | 0.377 | 0.348 | 0.379 | **0.342** | 0.373 | **0.342** | 0.373 |
| ETTm2 | 96 | 0.161 | 0.232 | 0.160 | **0.231** | 0.159 | 0.232 | **0.157** | **0.231** | 0.161 | 0.254 | **0.160** | 0.251 | 0.161 | **0.248** | **0.160** | **0.248** |
| | 192 | 0.227 | 0.277 | 0.225 | 0.276 | 0.223 | 0.276 | **0.219** | **0.274** | 0.213 | 0.290 | 0.213 | 0.289 | **0.212** | **0.286** | 0.213 | **0.286** |
| | 336 | 0.287 | 0.318 | 0.286 | 0.317 | 0.280 | 0.315 | **0.276** | **0.312** | **0.262** | 0.323 | 0.269 | 0.326 | 0.266 | **0.320** | 0.268 | 0.322 |
| | 720 | 0.383 | 0.379 | 0.381 | 0.377 | 0.368 | 0.372 | **0.363** | **0.369** | 0.341 | 0.373 | 0.346 | 0.379 | 0.357 | 0.379 | **0.338** | **0.369** |
| | **Avg.** | 0.265 | 0.302 | 0.263 | 0.300 | 0.258 | 0.299 | **0.254** | **0.297** | **0.244** | 0.310 | 0.247 | 0.311 | 0.249 | 0.308 | 0.245 | 0.306 |
| Weather | 96 | 0.156 | 0.190 | 0.174 | 0.209 | 0.152 | 0.191 | **0.141** | **0.175** | 0.153 | 0.208 | 0.150 | 0.202 | **0.145** | **0.195** | **0.145** | **0.195** |
| | 192 | 0.198 | 0.233 | 0.231 | 0.258 | 0.203 | 0.239 | **0.186** | **0.220** | 0.192 | 0.245 | 0.192 | 0.243 | 0.189 | 0.241 | **0.188** | **0.240** |
| | 336 | 0.248 | 0.272 | 0.289 | 0.298 | 0.267 | 0.283 | **0.239** | **0.262** | 0.240 | 0.281 | 0.237 | 0.279 | 0.238 | 0.278 | **0.236** | **0.277** |
| | 720 | 0.324 | 0.323 | 0.376 | 0.347 | 0.330 | 0.329 | **0.319** | **0.315** | **0.301** | **0.323** | 0.314 | 0.335 | 0.314 | 0.338 | 0.313 | 0.335 |
| | **Avg.** | 0.232 | 0.255 | 0.268 | 0.278 | 0.238 | 0.261 | **0.221** | **0.243** | 0.222 | 0.264 | 0.223 | 0.265 | 0.222 | 0.263 | **0.221** | **0.262** |

transformation matrix, instead of the original ones. Similar to the performance gap between full fine-tuning and LoRA over the original TSFMs, applying full fine-tuning to pruned TSFMs is usually better than applying LoRA to pruned TSFMs. As for comparisons between LoRA and Prune+LoRA, the pruned ones can yield better performance except for a few cases. It is noteworthy that most GPU memory overhead during time series forecasting is caused by self-attention computations over long contexts, rather than the model parameters and the optimizer. Since the model sizes of existing TSFMs are much smaller than LLMs, parameter-efficient fine-tuning methods do not significantly reduce memory. Thus, we would like to adopt full fine-tuning for better performance.

## C.4 Performance of Variant Pruning Methods

In light of the inherent sparsity of TSFMs, we introduce a variant of TSFM pruning, denoted as Prune$_{stat}$. This method prunes TSFMs based on statistics during the forward pass, specifically including the relative output norm of attention heads and the activation probability of FFN channels. We collect these statistics over the whole training set and subsequently remove insignificant heads and channels whose statistics fall below a predefined threshold. The threshold for relative output norm is selected from $\{0\%, 0.5\%, 1\%, 2\%\}$, while that for activation probabilities is selected from $\{0\%, 1\%, 2\%, 5\%\}$. As a counterpart, the pruning method based on importance scores is denoted as Prune$_{imp}$.

Table 9: Performance comparison between different pruning methods. $H$ denotes the forecast horizon. "FT" is short for fine-tuning.

| Models | | TimeMoE$_{base}$ | | | | | | TimesFM$_{2.0\text{-}500M}$ | | | | | | Chronos$_{base}$ | | | | | | TTM$_A$ | | | | | |
|---|---|---|---|---|---|---|---|---|---|---|---|---|---|---|---|---|---|---|---|---|---|---|---|---|---|
| Methods | | FT | | Prune$_{stat}$+FT | | Prune$_{imp}$+FT | | FT | | Prune$_{stat}$+FT | | Prune$_{imp}$+FT | | FT | | Prune$_{stat}$+FT | | Prune$_{imp}$+FT | | FT | | Prune$_{stat}$+FT | | Prune$_{imp}$+FT | |
| Data | H | MSE | MAE | MSE | MAE | MSE | MAE | MSE | MAE | MSE | MAE | MSE | MAE | MSE | MAE | MSE | MAE | MSE | MAE | MSE | MAE | MSE | MAE | MSE | MAE |
| ETTh1 | 96 | 0.340 | 0.377 | **0.339** | 0.374 | 0.340 | 0.376 | 0.384 | **0.391** | 0.385 | **0.391** | **0.383** | 0.393 | 0.366 | **0.374** | 0.366 | **0.374** | 0.366 | 0.375 | 0.359 | 0.395 | 0.359 | 0.395 | **0.358** | **0.393** |
| | 192 | 0.380 | 0.408 | **0.374** | **0.400** | 0.376 | 0.404 | 0.417 | **0.412** | 0.419 | **0.412** | **0.414** | 0.413 | 0.418 | 0.405 | 0.418 | 0.405 | **0.417** | 0.405 | 0.392 | 0.417 | **0.390** | 0.414 | 0.391 | **0.410** |
| | 336 | 0.406 | 0.432 | **0.393** | **0.415** | 0.399 | 0.425 | 0.430 | **0.422** | 0.433 | 0.423 | **0.419** | **0.422** | 0.453 | 0.424 | 0.453 | 0.424 | **0.449** | **0.422** | 0.407 | 0.430 | 0.408 | 0.431 | **0.402** | **0.424** |
| | 720 | 0.437 | 0.469 | **0.406** | **0.440** | 0.424 | 0.457 | 0.432 | **0.441** | 0.434 | **0.441** | **0.429** | 0.444 | 0.452 | 0.441 | 0.451 | 0.437 | 0.451 | 0.437 | 0.448 | 0.467 | **0.437** | **0.461** | 0.449 | 0.465 |
| | Avg. | 0.391 | 0.422 | **0.378** | **0.407** | 0.385 | 0.416 | 0.416 | **0.417** | 0.418 | **0.417** | **0.417** | 0.418 | 0.422 | 0.411 | 0.422 | **0.410** | 0.421 | **0.410** | 0.402 | 0.427 | **0.399** | 0.425 | 0.400 | **0.423** |
| ETTh2 | 96 | 0.267 | 0.330 | 0.283 | 0.345 | **0.262** | **0.327** | 0.268 | 0.324 | **0.267** | **0.323** | 0.268 | 0.325 | 0.266 | **0.311** | 0.264 | **0.311** | 0.266 | **0.311** | 0.264 | 0.333 | 0.263 | 0.332 | **0.262** | **0.328** |
| | 192 | 0.341 | 0.380 | 0.366 | 0.403 | **0.331** | **0.374** | 0.329 | 0.367 | **0.327** | **0.364** | 0.330 | 0.368 | 0.334 | 0.357 | **0.329** | **0.355** | 0.331 | 0.356 | **0.321** | **0.372** | **0.321** | **0.372** | 0.324 | **0.370** |
| | 336 | 0.386 | 0.415 | 0.429 | 0.448 | **0.372** | **0.403** | 0.361 | 0.391 | **0.359** | **0.389** | 0.360 | 0.392 | 0.378 | 0.391 | **0.367** | **0.387** | 0.370 | 0.388 | 0.351 | 0.397 | **0.350** | 0.397 | 0.351 | **0.396** |
| | 720 | 0.419 | 0.454 | 0.559 | 0.530 | **0.377** | **0.423** | 0.384 | 0.413 | **0.379** | **0.410** | 0.383 | 0.414 | 0.395 | 0.418 | **0.384** | **0.413** | 0.389 | 0.415 | 0.395 | 0.439 | 0.395 | 0.439 | **0.393** | **0.437** |
| | Avg. | 0.353 | 0.395 | 0.409 | 0.432 | **0.336** | **0.382** | 0.336 | 0.374 | **0.333** | **0.372** | 0.335 | 0.375 | 0.343 | 0.369 | **0.336** | **0.367** | 0.339 | 0.368 | 0.333 | 0.385 | **0.332** | 0.385 | 0.333 | **0.383** |
| ETTm1 | 96 | 0.220 | 0.302 | 0.221 | 0.301 | 0.220 | 0.302 | 0.321 | 0.338 | 0.316 | **0.336** | **0.298** | 0.338 | 0.285 | 0.315 | 0.285 | 0.315 | **0.283** | 0.315 | 0.295 | 0.346 | 0.298 | 0.347 | **0.291** | **0.340** |
| | 192 | 0.284 | 0.347 | **0.278** | 0.343 | 0.282 | 0.344 | 0.366 | 0.365 | 0.359 | **0.363** | **0.340** | 0.365 | 0.329 | 0.345 | 0.329 | **0.344** | 0.326 | **0.344** | 0.332 | 0.366 | 0.330 | 0.366 | **0.327** | **0.363** |
| | 336 | 0.349 | 0.392 | **0.333** | 0.382 | 0.335 | **0.381** | 0.390 | 0.385 | 0.382 | **0.382** | **0.368** | 0.385 | 0.361 | 0.369 | 0.361 | 0.368 | **0.356** | **0.367** | 0.356 | 0.383 | 0.358 | 0.384 | **0.356** | **0.381** |
| | 720 | 0.462 | 0.466 | 0.427 | 0.444 | **0.415** | **0.437** | 0.434 | 0.415 | 0.423 | **0.411** | **0.419** | 0.420 | 0.434 | 0.413 | 0.430 | 0.409 | **0.423** | **0.407** | 0.409 | 0.420 | **0.392** | **0.405** | 0.395 | 0.406 |
| | Avg. | 0.329 | 0.377 | 0.315 | 0.368 | **0.313** | **0.366** | 0.378 | 0.376 | 0.370 | **0.373** | **0.356** | 0.377 | 0.352 | 0.361 | 0.351 | 0.359 | **0.347** | **0.358** | 0.348 | 0.379 | 0.345 | 0.376 | **0.342** | **0.373** |
| ETTm2 | 96 | 0.161 | 0.253 | 0.161 | 0.250 | **0.158** | **0.251** | 0.169 | 0.250 | 0.168 | **0.249** | **0.166** | 0.251 | 0.160 | **0.231** | 0.160 | 0.233 | **0.157** | **0.231** | 0.160 | 0.251 | **0.159** | 0.251 | 0.160 | **0.248** |
| | 192 | 0.229 | 0.308 | 0.226 | 0.299 | **0.217** | **0.299** | 0.232 | 0.295 | 0.229 | **0.293** | **0.225** | 0.294 | 0.225 | 0.276 | 0.226 | 0.278 | **0.219** | **0.274** | 0.213 | 0.289 | 0.215 | 0.294 | **0.213** | **0.286** |
| | 336 | 0.313 | 0.367 | 0.305 | 0.353 | **0.273** | **0.342** | 0.291 | 0.335 | 0.287 | 0.334 | **0.279** | **0.332** | 0.286 | 0.317 | 0.287 | 0.318 | **0.276** | **0.312** | 0.269 | 0.326 | 0.271 | 0.332 | **0.268** | **0.322** |
| | 720 | 0.486 | 0.464 | 0.471 | 0.449 | **0.375** | **0.407** | 0.367 | 0.389 | 0.368 | 0.390 | **0.358** | **0.387** | 0.381 | 0.377 | 0.381 | 0.377 | **0.363** | **0.369** | 0.346 | 0.379 | 0.340 | 0.379 | **0.338** | **0.369** |
| | Avg. | 0.297 | 0.348 | 0.291 | 0.338 | **0.256** | **0.325** | 0.265 | 0.317 | 0.263 | 0.317 | **0.257** | **0.316** | 0.263 | 0.300 | 0.264 | 0.302 | **0.254** | **0.297** | 0.247 | 0.311 | 0.246 | 0.314 | **0.245** | **0.306** |
| Weather | 96 | 0.141 | **0.194** | 0.141 | **0.194** | 0.141 | 0.196 | - | - | - | - | - | - | 0.174 | 0.209 | 0.159 | 0.202 | **0.141** | **0.175** | 0.150 | 0.202 | 0.149 | 0.200 | **0.145** | **0.195** |
| | 192 | 0.196 | 0.254 | 0.191 | **0.249** | 0.191 | 0.250 | - | - | - | - | - | - | 0.231 | 0.258 | 0.210 | 0.249 | **0.186** | **0.220** | 0.192 | 0.243 | 0.194 | 0.245 | **0.188** | **0.240** |
| | 336 | 0.260 | 0.309 | 0.251 | 0.302 | **0.249** | **0.301** | - | - | - | - | - | - | 0.289 | 0.298 | 0.265 | 0.289 | **0.239** | **0.262** | 0.237 | 0.279 | **0.236** | 0.276 | 0.236 | 0.277 |
| | 720 | 0.346 | 0.378 | 0.347 | 0.380 | **0.331** | **0.370** | - | - | - | - | - | - | 0.376 | 0.347 | 0.362 | 0.344 | **0.319** | **0.315** | 0.314 | 0.335 | **0.304** | **0.326** | 0.313 | 0.335 |
| | Avg. | 0.236 | 0.284 | 0.233 | 0.281 | **0.228** | **0.279** | - | - | - | - | - | - | 0.268 | 0.278 | 0.249 | 0.271 | **0.221** | **0.243** | 0.223 | 0.265 | 0.221 | 0.262 | **0.221** | **0.262** |
| Traffic | 96 | - | - | - | - | - | - | - | - | - | - | - | - | 0.335 | 0.214 | 0.333 | 0.212 | **0.332** | **0.211** | 0.355 | 0.259 | **0.355** | 0.259 | 0.356 | 0.259 |
| | 192 | - | - | - | - | - | - | - | - | - | - | - | - | 0.352 | 0.223 | 0.351 | 0.221 | **0.350** | **0.220** | 0.368 | 0.263 | 0.369 | 0.263 | 0.371 | 0.265 |
| | 336 | - | - | - | - | - | - | - | - | - | - | - | - | 0.377 | **0.235** | 0.375 | 0.233 | **0.374** | **0.231** | 0.387 | 0.273 | 0.385 | 0.273 | **0.385** | 0.273 |
| | 720 | - | - | - | - | - | - | - | - | - | - | - | - | 0.430 | 0.260 | 0.426 | 0.259 | **0.426** | **0.257** | 0.438 | 0.306 | 0.431 | 0.296 | **0.428** | **0.294** |
| | Avg. | - | - | - | - | - | - | - | - | - | - | - | - | 0.374 | 0.233 | **0.371** | 0.231 | **0.371** | **0.230** | 0.387 | 0.275 | **0.385** | 0.273 | **0.385** | 0.273 |

\* The Weather and Traffic datasets included in the pre-training are not evaluated, denoted by a dash (–).

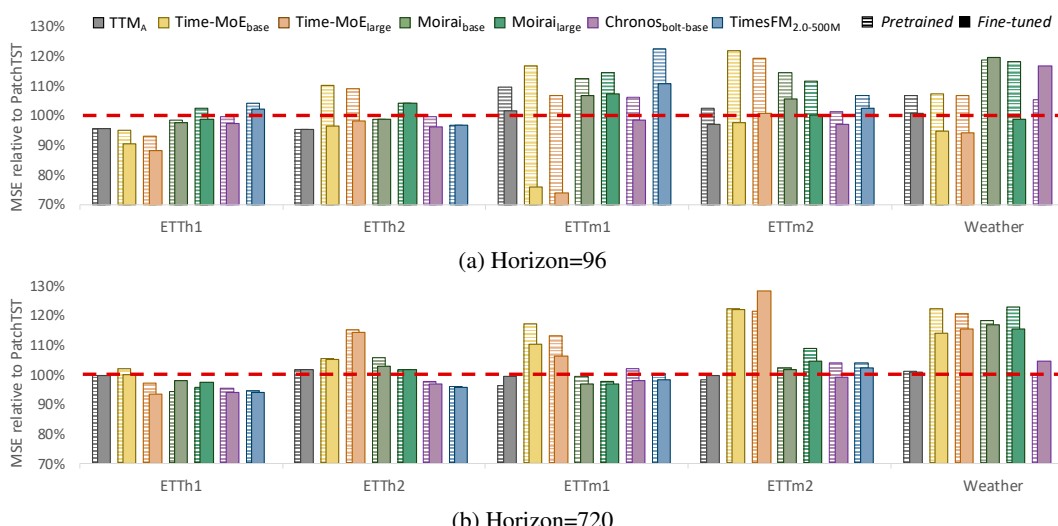

(a) Horizon=96

(b) Horizon=720

Fig. 10: MSE of TSFMs and PatchTST when forecasting 96 and 720 steps.

Table 9 presents forecasting performance of TSFMs specialized by variant pruning methods and full fine-tuning. The method "Prune$_{stat}$+FT" usually achieves better performance than fine-tuning, which confirms the effectiveness of architecture specialization. Nevertheless, "Prune$_{stat}$+FT" cannot outperform "Prune$_{imp}$+FT" in most cases, indicating that only pruning attention heads or intermediate FFN channels is not sufficient for specialization. In a few cases, "Prune$_{imp}$+FT" underperforms "Prune$_{stat}$+FT", e.g., given Time-MoE$_{base}$ on ETTh1 where "Prune$_{stat}$+FT" removes several attention heads while "Prune$_{imp}$+FT" primarily focus on FFNs. We conjecture that a more comprehensive pruning strategy, combining both statistical and importance score-based criteria, could potentially lead to better performance.

## C.5 Zero-Shot Performance

We provide comparisons between zero-shot performance and full-shot performance in Table 10.

# D   Limitations and Future Works

Our pruning metrics are calculated based on the training set, which may incur overfitting risks in extremely non-stationary time series data. To obtain a robust pruned model, possible solutions include data augmentation and adversarial attack techniques that keep the pruning metrics insensitive to noise. Furthermore, our pruning method works on TSFMs with strong zero-shot forecasting performance, since our basic assumption is that the prior knowledge of architecture specialization acquired by pre-training is effective for downstream forecasting. As for some tasks where pre-trained TSFMs show poor capabilities and cannot deliver an effective subnetwork, one possible solution is to apply pruning to a fine-tuned TSFM, rewind remaining parameters to the pre-trained weights, and fine-tune the pruned model. This follows prior works based on the classic Lottery Ticket Hypothesis [8], while we would like to leave it as a future work.

Table 10: Full results of forecasting performance. "FT" is short for fine-tuning. Results of PatchTST are obtained from Li et al.. A lower MSE or MAE indicates a better prediction. For each comparison between results without and with pruning, we mark the winner in **bold**. For each prediction task, we mark the best results in yellow.

| Models | | TimeMoE$_{base}$ | | | | | | TimeMoE$_{large}$ | | | | | | TimerXL | | | | | | Moirai$_{base}$ | | | | | | Moirai$_{large}$ | | | | | | Chronos$_{base}$ | | | | | | TTM$_A$ | | | | | PatchTST |
|---|---|---|---|---|---|---|---|---|---|---|---|---|---|---|---|---|---|---|---|---|---|---|---|---|---|---|---|---|---|---|---|---|---|---|---|---|---|---|---|---|---|---|---|
| Methods | | Zero-shot | | FT | | Prune+FT | | Zero-shot | | FT | | Prune+FT | | Zero-shot | | FT | | Prune+FT | | Zero-shot | | FT | | Prune+FT | | Zero-shot | | FT | | Prune+FT | | Zero-shot | | FT | | Prune+FT | | Zero-shot | | FT | | Prune+FT | | |
| Datasets | H | MSE | MAE | MSE | MAE | MSE | MAE | MSE | MAE | MSE | MAE | MSE | MAE | MSE | MAE | MSE | MAE | MSE | MAE | MSE | MAE | MSE | MAE | MSE | MAE | MSE | MAE | MSE | MAE | MSE | MAE | MSE | MAE | MSE | MAE | MSE | MAE | MSE | MAE | MSE | MAE | MSE | MAE | MSE | MAE |

| Datasets | H | TimeMoE$_{base}$ | | | | | | TimeMoE$_{large}$ | | | | | | TimerXL | | | | | | Moirai$_{base}$ | | | | | | Moirai$_{large}$ | | | | | | Chronos$_{base}$ | | | | | | TTM$_A$ | | | | | | PatchTST |
|---|---|---|---|---|---|---|---|---|---|---|---|---|---|---|---|---|---|---|---|---|---|---|---|---|---|---|---|---|---|---|---|---|---|---|---|---|---|---|---|---|---|---|---|---|
| Win Count | | 5 | | 13 | | 19 | 15 | 5 | | 13 | | 14 | 12 | 5 | | 3 | | 4 | 9 | 9 | | 5 | | 8 | 7 | 5 | | 3 | | 8 | 7 | 21 | | 27 | | 23 | 25 | 29 | | 30 | | 13 | 30 | 27 |
| Win Rate | | 8.3% | | 46.0% | | 68.0% | | 8.3% | | 46.0% | | 52.0% | | 10.0% | | 11.7% | | 26.7% | | 23.3% | | 25.0% | | 33.3% | | 10.0% | | 18.3% | | 20.0% | | 80.0% | | 80.0% | | 98.3% | | 71.7% | | 85.0% | 95.0% | |