# OpenReview forum: "Less is More: Unlocking Specialization of Time Series Foundation Models via Structured Pruning"
_NeurIPS.cc/2025/Conference — NeurIPS 2025 poster_

### Official Review · Reviewer_SzHa · 2025-06-29

**Clarity:** 3
**Significance:** 2
**Originality:** 2
**Rating:** 3
**Confidence:** 5

**Summary:**

The paper attempts to address an important problem in the time series forecasting domain: How to prune the pre-trained models during Full-Shot fine-tuning (referring to usage of 100% of the downstream data for fine-tuning). The paper proposes a pruning and then fine-tune approach, which essentially masks some of the neural network connections and thus create a more compact model. The paper evaluates the proposed approach in six time series forecasting datasets which are widely employed in this domain.

**Questions:**

1.	Undoubtedly, the main application of time series foundation models (TSFMs) are in the zero-shot or few-shot settings. Hence, the same experiments and analysis should be done in few-shot fine-tuning settings with different percentages of training data employed (20%, 40%, …, 100%). Then only, the efficacy of the proposed approach will be clear.
2.	The pruning method is intuitively similar to dropouts as it essentially drops some network connections. Is that correct? Have the author(s) tried to apply varying dropout (best value chosen using a validation set) during the fine-tuning. A full comparison with dropouts should be performed for all models and datasets.
3.	Some of the TSFMs suggest fine-tuning only the model’s head for the downstream task (e.g., the HeadProbing in TTM). How does the pruning method perform when only applied on the head (or other layers for other TSFMs)?
4.	The improvement in the MSE and MAE scores are very insignificant, often 2nd and 3rd decimal places. This questions the efficacy of the proposed method.
5.	Only three unique data sources have been used (ETT, electricity, weather). More variety of datasets should be included in the analysis, e.g, the traffic dataset widely used for evaluation TSFMs.

**Ethical Concerns:**

["NO or VERY MINOR ethics concerns only"]

**Final Justification:**

While the rebuttal has clarified many of my concerns, I still believe the evaluation is weak as I mentioned in my replies. The proposed method of pruning is model agnostic, which is good for its applicability, but the evaluation setting and reported metrics limit the understanding on how much gain is possible with the help of pruning.

**Limitations:**

No.
Limitations of the proposed method should be discussed. See the weaknesses section for details. A summary is provided here:
1. How does the method perform in few-shot setting compared to a model trained from scratch with few-shot data as well?
2. Comparison with HPO-selected dropouts.
3. Comparison when only head is pruned.
4. Wider variety of data sources.
5. Insignificant improvements.

**Quality:**

2

**Strengths And Weaknesses:**

Strengths:

1.	The problem is pretty important in the time series domain, where, often fine-tuning cannot match the performance of a model trained from scratch on the entire data.
2.	Rigorous experimental evidence is shown for existing SOTA models that possess the addressed problem.
3.	Minimal necessary ablation studies have been done.

Weaknesses:

1.	Undoubtedly, the main application of time series foundation models (TSFMs) are in the zero-shot or few-shot settings. Hence, the same experiments and analysis should be done in few-shot fine-tuning settings with different percentages of training data employed (20%, 40%, …, 100%). Then only, the efficacy of the proposed approach will be clear.
2.	The pruning method is intuitively similar to dropouts as it essentially drops some network connections. Is that correct? Have the author(s) tried to apply varying dropout (best value chosen using a validation set) during the fine-tuning. A full comparison with dropouts should be performed for all models and datasets.
3.	Some of the TSFMs suggest fine-tuning only the model’s head for the downstream task (e.g., the HeadProbing in TTM). How does the pruning method perform when only applied on the head (or other layers for other TSFMs)?
4.	The improvement in the MSE and MAE scores are very insignificant, often 2nd and 3rd decimal places. This questions the efficacy of the proposed method.
5.	Only three unique data sources have been used (ETT, electricity, weather). More variety of datasets should be included in the analysis, e.g, the traffic dataset widely used for evaluation TSFMs.

---

> ### Author Rebuttal · Authors · 2025-07-31
>
> **Q5: More variety of datasets should be included**
>
> Thanks for your advice! We additionally perform short-term forecasting over Traffic and the M4 Yearly and Covid Death benchmark borrowed from GIFT-EVAL. The results of Chronos-bolt-base are reported in the following table. Our proposed method outperforms the fine-tuning baseline on almost all the metric.
>
> |                                          | Chronos-bolt-base   | MSE         | MAE     | MASE      | CRPS      | NRMSE     |
> | ---------------------------------------- | ------------------- | ----------- | ------- | --------- | --------- | --------- |
> | M4 Yearly (context length$\in[19, 284]$) | Zero-shot           | 3926423     | 943     | 3.507     | 0.121     | 0.318     |
> |                                          | Fine-Tune           | 3226941     | 833     | 3.089     | 0.107     | 0.309     |
> |                                          | Prune-then-Finetune | **3125398** | **816** | **3.007** | **0.104** | **0.283** |
> | Covid Death (context length=212)         | Zero-shot           | 641888      | 164     | 38.9      | 0.047     | 0.301     |
> |                                          | Fine-Tune           | 577110      | 157     | **37.5**  | 0.046     | 0.286     |
> |                                          | Prune-then-Finetune | **239526**  | **93**  | 49.8      | **0.034** | **0.184** |
>
> **Q1: Undoubtedly, the main application of time series foundation models (TSFMs) are in the zero-shot or few-shot settings. Hence, the same experiments and analysis should be done in few-shot fine-tuning settings with different percentages of training data employed (20%, 40%, …, 100%). Then only, the efficacy of the proposed approach will be clear.**
>
> As shown in the table above, fine-tuning significantly outperforms zero-shot forecasting, and our proposed approach delivers considerable performance gains. We believe that it is time to improve full-shot performance of TSFMs for wider applications.
>
> As for few-shot performance (with 20% training data), the following tables show the average MSE and MAE with the horizon in {96, 192, 336, 720}. FT is short for Fine-tuning, and Prune+FT is short for our proposed Prune-then-Finetune.
>
> |          | Chronos     | -bolt | -base     |           |             | TTM   |           |           |
> | -------- | ----------- | ----- | --------- | --------- | ----------- | ----- | --------- | --------- |
> |          | **FT** |       | **Prune** | **+FT**   | **FT** |       | **Prune** | **+FT**   |
> | Datasets | MSE         | MAE   | MSE       | MAE       | MSE         | MAE   | MSE       | MAE       |
> | ETTh1    | 0.430       | 0.414 | **0.428** | **0.413** | 0.408       | 0.432 | **0.402** | **0.426** |
> | ETTh2    | 0.343       | 0.369 | **0.339** | **0.368** | 0.333       | 0.385 | 0.333     | **0.383** |
> | ETTm1    | 0.359       | 0.359 | **0.350** | **0.358** | 0.350       | 0.379 | **0.342** | **0.373** |
> | ETTm2    | 0.271       | 0.305 | **0.254** | **0.297** | 0.245       | 0.310 | **0.244** | **0.306** |
> | Weather  | 0.248       | 0.269 | **0.221** | **0.263** | 0.225       | 0.267 | **0.221** | **0.263** |
> | ECL      | 0.154       | 0.240 | **0.153** | **0.239** | 0.167       | 0.265 | **0.165** | **0.263** |
>
> As shown in the table, our method can still outperforms fine-tuning all parameters in few-shot scenarios.
>
> **Q2: The pruning method is intuitively similar to dropouts as it essentially drops some network connections. Is that correct? A full comparison with dropouts should be performed for all models and datasets.**
>
> Thanks for your question! Pruning and dropout are essentially different. Dropouts randomly corrupt hidden states during training, while the inference time involves the complete network without droping any connections. By contrast, connections are dropped permanently after pruning, and we consistently use the same subnetwork during finetuning and inference.
>
> We compare the average MSE and MAE of Chronos-bolt-base without and with dropout in the following table.
>
> | Methods  | Fine-Tune | (dropout=0) | Fine-Tune | (dropout=0.1) |
> | -------- | --------- | ----------- | --------- | ------------- |
> | Datasets | MSE       | MAE         | MSE       | MAE           |
> | ETTh1    | **0.420** | **0.410**   | 0.422     | 0.411         |
> | ETTh2    | **0.339** | **0.368**   | 0.343     | 0.369         |
> | ETTm1    | **0.345** | **0.357**   | 0.352     | 0.361         |
> | ETTm2    | **0.255** | **0.297**   | 0.263     | 0.300         |
> | Weather  | **0.261** | **0.268**   | 0.268     | 0.278         |
> | ECL      | 0.154     | 0.240       | **0.153** | **0.239**     |
>
> The results demonstrate that dropout can result in negative effects. We assume that the dropout technique is not suitable for time series. In computer vision tasks, it is naturally required and relatively easy to conduct classification or reconstruction even with noisy pixels. By contrast, it is not suitable to always consider all latent features in time series, which may contain noisy fluctuations or outdated information.
>
> **Q3: Some of the TSFMs suggest fine-tuning only the model’s head (e.g., the HeadProbing in TTM)**
>
> Thanks for your advice! In the following tables, we compare performance of probing and full finetuning in terms of average MSE and MAE with the horizon in {96, 192, 336, 720}. The dropout hyperparameter has been tuned based on the validation set.
>
> Chronos-bolt-base:
>
> | Methods  | Probing |       | Full      | Finetune  | Prune-then- | Finetune  |
> | -------- | ------- | ----- | --------- | --------- | ----------- | --------- |
> | Datasets | MSE     | MAE   | MSE       | MAE       | MSE         | MAE       |
> | ETTh1    | 0.420   | 0.411 | 0.420     | **0.410** | **0.417**   | **0.410** |
> | ETTh2    | 0.348   | 0.373 | **0.339** | **0.368** | **0.339**   | **0.368** |
> | ETTm1    | 0.356   | 0.360 | 0.345     | 0.357     | **0.344**   | **0.356** |
> | ETTm2    | 0.252   | 0.301 | 0.255     | 0.297     | **0.247**   | **0.296** |
> | Weather  | 0.226   | 0.250 | 0.261     | 0.268     | **0.215**   | **0.243** |
> | ECL      | 0.152   | 0.238 | 0.154     | 0.240     | **0.150**   | **0.237** |
>
> TTM:
>
> | Methods  | Probing |       | Full  | Finetune | Prune-then- | Finetune  |
> | -------- | ------- | ----- | ----- | -------- | ----------- | --------- |
> | Datasets | MSE     | MAE   | MSE   | MAE      | MSE         | MAE       |
> | ETTh1    | 0.403   | 0.424 | 0.402 | 0.427    | **0.399**   | **0.423** |
> | ETTh2    | 0.333   | 0.385 | 0.333 | 0.385    | 0.333       | **0.383** |
> | ETTm1    | 0.353   | 0.377 | 0.348 | 0.379    | **0.342**   | **0.373** |
> | ETTm2    | 0.252   | 0.307 | 0.247 | 0.311    | **0.245**   | **0.306** |
> | Weather  | 0.225   | 0.264 | 0.223 | 0.265    | **0.221**   | **0.262** |
> | ECL      | 0.162   | 0.261 | 0.161 | 0.258    | **0.160**   | **0.256** |
>
> The results demonstrate that only fine-tuning the final head does not exhibit significant performance improvement. In a few cases, head probing outperforms full finetuning, while our proposed prune-then-finetune method still achieves the best performance.
>
> As for TTM, their [official code](https://github.com/ibm-granite/granite-tsfm/blob/63c962fee413ed53789ea0d16119218d09f28d1a/notebooks/hfdemo/tinytimemixer/full_benchmarking/gift_leaderboard/ttm.ipynb) suggests finetuning full parameters to achieve higher performance on benchmarks including [GIFT-EVAL](huggingface.co/datasets/Salesforce/GiftEval/). We would like to consider head probing as an approach to training efficiency.

---

> ### Author Response · Authors · 2025-08-02
> **Additional Experimental Results**
>
> **Q5: More variety of datasets should be included**
>
> We have experimented on more benchmarks, including the Traffic benchmark and 8 datasets from GIFT-Eval, and we would like to share our new results here.
>
> Additional results of long-term forecasting:
> |||Chronos||||||||||TTM||||
> |-|-|-|-|-|-|-|-|-|-|-|-|-|-|-|-|
> ||| **Fine-Tune** |  |  | **Prune+FT** |  |  |  |  |  | **Fine-Tune** |  |  | **Prune+FT** |  |
> |**Dataset** | **Horizon** | **MSE** | **MAE** |  | **MSE** | **MAE** |  |  |  |  | **MSE** | **MAE** |  | **MSE** | **MAE** |
> |traffic| 96 | 0.335 | 0.214 |  | **0.332** | **0.211** |  |  |  |  | **0.355** | 0.259 |  | 0.356 | 0.259 |
> ||192| 0.352 | 0.223 |  | **0.350** | **0.220** |  |  |  |  | **0.368** | **0.263** |  | 0.371 | 0.265 |
> ||336| 0.377 | 0.235 |  | **0.374** | **0.231** |  |  |  |  | 0.387 | 0.273 |  | **0.385** | 0.273 |
> ||720| 0.430 | 0.260 |  | **0.426** | **0.257** |  |  |  |  | 0.438 | 0.306 |  | **0.428** | **0.294** |
> ||Avg.| 0.374 | 0.233 |  | **0.371** | **0.230** |  |  |  |  | 0.387 | 0.275 |  | **0.385** | **0.273** |
>
> Additional results of short-term forecasting are reported in the following table. Metrics are calculated by following GIFT-Eval, e.g., MSE and MAE are based on raw values without z-score normalization.
>
> ||Chronos-bolt-base|MSE|MAE|MASE|CRPS|NRMSE|
> |-|-|-|-|-|-|-|
> |M4 Quarterly|Zero-shot|1878597|578|1.224|0.077|0.229|
> ||Fine-Tune|1775734|571|**1.221**|0.075|0.223|
> ||Prune-then-Finetune|**1749123**|**551**|1.225|**0.074**|**0.221**|
> |M4 Monthly|Zero-shot|1934416|565|0.949|0.094|0.289|
> ||Fine-Tune|1894232|571|0.946|0.095|0.286|
> ||Prune-then-Finetune|**1820581**|**556**|**0.920**|**0.091**|**0.280**|
> |M4 Hourly|Zero-shot|1154587|244|0.837|0.025|0.147|
> ||Fine-Tune|1032867|239|0.943|0.026|0.139|
> ||Prune-then-Finetune|**937494**|**227**|**0.805**|**0.024**|**0.131**|
> |us_births/D|Zero-shot|261850|329|0.485|0.026|0.048|
> ||Fine-Tune|171933|248|0.365|0.019|0.039|
> ||Prune-then-Finetune|**168877**|**241**|**0.354**|**0.018**|**0.039**|
> |saugeen/D|Zero-shot|1082|12.8|2.84|0.338|1.065|
> ||Fine-Tune|1057|13.0|2.889|0.351|1.053|
> ||Prune-then-Finetune|**1052**|**11.9**|**2.642**|**0.328**|**1.051**|
> |car_parts|Zero-shot|1.39|0.48|0.855|0.995|2.827|
> ||Fine-Tune|1.378|0.438|0.835|0.916|2.815|
> ||Prune-then-Finetune|**1.365**|**0.435**|**0.826**|**0.910**|**2.802**|
> |restaurant|Zero-shot|146.6|7.329|0.700|0.264|0.557|
> ||Fine-Tune|143.8|7.192|0.687|0.259|0.552|
> ||Prune-then-Finetune|**142.5**|**7.159**|**0.684**|**0.258**|**0.549**|
>
> Note that TTM sets a short context length and thereby a lightweight MLP Mixer network for short time series including M4 Yearly, M4 Quarterly, M4 Monthly, car_parts and restaurant. Applying pruning to this tiny model, we observe a performance decline on the validation set, i.e., the optimal pruning ratio after hyperparameter selection is 0. Thus, we only report the performance of TTM on other benchmarks as follows:
>
> ||TTM|MSE|MAE|MASE|CRPS|NRMSE|
> |-|-|-|-|-|-|-|
> |Covid Death|Zero-shot|2181495|327|53.5|0.123|0.56|
> ||Fine-Tune|688993|125|30.8|0.047|0.361|
> ||Prune-then-Finetune|**584696**|**111**|**29.2**|**0.042**|**0.287**|
> |M4 Hourly|Zero-shot|**2191172**|**294**|2.783|**0.040**|**0.202**|
> ||Fine-Tune|2203642|300|1.029|0.035|0.203|
> ||Prune-then-Finetune|2258110|**294**|**0.817**|**0.040**|0.205|
> |us_births/D|Zero-shot|1462623|1106|1.629|0.104|0.113|
> ||Fine-Tune|205838|282|0.415|0.026|0.043|
> ||Prune-then-Finetune|**145496**|**239**|**0.354**|**0.022**|**0.035**|
> |saugeen/D|Zero-shot|1256|18.2|4.034|0.589|1.148|
> ||Fine-Tune|1258|13.4|3.000|0.406|1.149|
> ||Prune-then-Finetune|**855**|**12.5**|**2.782**|**0.394**|**0.947**|
>
> From the above results, we can see that our proposed method generalize well on these diverse data and usually outperforms the finetuning baseline, especially for Chronos.
>
> **Q4: Insignificant improvement in the MSE and MAE scores.**
>
> The reported MSE and MAE over ETT, Electricity, Weather, and Traffic benchmarks are calculated based on normalized values. Small improvements can translate to meaningful performance gains with respect to the original data quantity. For those datasets from GIFT-Eval, we calculate MSE and MAE based on raw values, and we can see that the improvements are significant. Undoubtedly, it is meaningful to downstream decision-making by improving the full-shot performance.
>
> Moreover, our method shows consistent improvements across a wide range of benchmarks and settings, indicating strong robustness rather than performance gains limited to a few cases. In addition, the pruning process leads to a reduced model size and computation cost, making our method more efficient without sacrificing accuracy, which can benefit real-world deployment.

---

> ### Author Response · Authors · 2025-08-05
> **Extended Few-Shot Results for Q1**
>
> **Q1: Few-shot settings**
>
> Dear Reviewer,
>
> Apart from previous experiments using 20% training data, we have performed more experiments with 40% training data as you wanted.
>
> **40% training data:**
>
> |  | Chronos |  |  |  |  |  |  |  |  |  | TTM |  |  |  |
> |---|---|---|---|---|---|---|---|---|---|---|---|---|---|---|
> |  | **Fine-Tune** |  |  | **Prune+FT** |  |  |  |  |  | **Fine-Tune** |  |  | **Prune+FT** |  |
> | **Datasets** | **MSE** | **MAE** |  | **MSE** | **MAE** |  |  |  |  | **MSE** | **MAE** |  | **MSE** | **MAE** |
> | ETTh1 | 0.428 | 0.414 |  | **0.426** | **0.412** |  |  |  |  | 0.406 | 0.430 |  | **0.402** | **0.425** |
> | ETTh2 | 0.344 | 0.370 |  | **0.341** | **0.368** |  |  |  |  | 0.334 | 0.387 |  | **0.333** | **0.383** |
> | ETTm1 | 0.358 | 0.359 |  | **0.349** | **0.358** |  |  |  |  | 0.350 | 0.380 |  | **0.342** | **0.374** |
> | ETTm2 | 0.269 | 0.304 |  | **0.251** | **0.298** |  |  |  |  | 0.246 | 0.311 |  | **0.243** | **0.305** |
> | Weather | 0.263 | 0.266 |  | **0.219** | **0.256** |  |  |  |  | 0.225 | 0.266 |  | **0.221** | **0.263** |
> | Electricity | 0.154 | 0.239 |  | **0.153** | **0.238** |  |  |  |  | 0.165 | 0.262 |  | **0.163** | **0.260** |
>
> The results confirm that our approach consistently generalize well. Also, it is worth mentioning we conduct experiments on smaller time series datasets, including M4 Hourly (414 samples), Covid Deaths (266 samples), Car Parts (2674 samples), Restaurant (807 samples), and US Births (7305 samples), where our approach is still effective in these few-shot forecasting tasks. Please refer to our 'Additional Experimental Results' in the previous comment.

---

> ### Author Response · Authors · 2025-08-05
> **Looking Forward to Your Feedback**
>
> Dear Reviewer SzHa,
>
> Thanks for your dedication to reviewing our paper!
>
> In our early response, we have included more experiments that demonstrate the effectiveness of our approach. We will highly appreciate it if you let us know whether your previous concerns have been adequately addressed. If you have any further questions, please do not hesitate to let us know, so that we can respond to them timely.

---

> > ### Comment · Reviewer_SzHa · 2025-08-05
> > **Feedback to rebuttal**
> >
> > Q5. What was the rationale behind selectively choosing the above datasets from GIFT (which has many more datasets)?
> >
> > Q2. Is Chronos-bolt-base suitable for the dropout experiment? Does it inherently support/recommend adding droupouts? TTM, on the other hand, supports dropout. Hence, this expt should be done with TTM or any other model that recommends dropouts.
> >
> > Q4. Can the authors calculate and report the average relative percentage improvement (maybe w.r.t. Table-2) for each dataset? I still see that the relative improvement is insignificant, which limit the potential of the proposed approach of model pruning. "Win Rate" might not be the ideal choice for comparison particularly when standard deviation of the metrics are not reported over multiple random seeds. The wins are happening at the 3rd decimal place for many entries.
> >
> > Q5 (TTM new expts): Can it be said that the pruning method mostly works on bigger models? Since, it did not help with TTM on short datasets, what was the rationale behind choosing those 4 new datasets for TTM's expt?
> >
> > _Limitations (missing from the draft)_: Related to the above queries, what are the limitations of the proposed technique? Does it work on small models? What is the relative percentage improvement?

---

> ### Author Response · Authors · 2025-08-07
>
> Thanks for your response! We have made more clarification to address your concerns.
>
> **Q5. Why choosing the above datasets**
>
> In addition to the suggested traffic dataset, we selected additional datasets from diverse domains, spanning healthcare, financial, traffic, sales, and nature, to demonstrate the robustness of our proposed solution.
>
> **Q2. Does Chronos inherently support droupout?**
>
> Chronos-bolt-base inherently supports using dropout, with a default dropout probability of 0.1 specified in their released configuration. Our Table 2 followed the dropout probabilities recommended by Chronos-bolt-base and TTM.
>
> It is important to note that dropout and structured pruning serve distinct purposes. Dropout randomly disables neurons during training, and the all neurons still serve for inference. In contrast, our structured pruning approach aims to identify and retain a task-relevant subnetwork, which is a kind of prior knowledge acquired by pretraining, bootstrapping targeted specialization for downstream tasks. This allows the model to focus its capacity on adapting parameters that are truly relevant to the target task.
>
> Unsurprisingly, our additional experiments with TTM show that dropout does not consistently improve fine-tuning performance, highlighting the need for our task-aware structured pruning method.
>
> |TTM|Fine-Tune|w/o dropout|Fine-Tune|w/ dropout|
> |-|-|-|-|-|
> |Datasets|MSE|MAE|MSE|MAE|
> |ETTh1|**0.401**|**0.425**|0.402|0.427|
> |ETTh2|0.333|0.386|0.333|**0.385**|
> |ETTm1|**0.344**|**0.375**|0.348|0.379|
> |ETTm2|0.250|**0.310**|**0.247**|0.311|
> |Weather|**0.221**|**0.263**|0.223|0.265|
> |ECL|0.163|0.259|**0.161**|**0.258**|
>
> **Q4. Relative improvement**
>
> The average relative improvements are listed in the following table.
>
> |ETTh1|ETTh2|ETTm1|ETTm2|Weather|Electricity|M4 Yearly|M4 Quarterly|M4 Monthly|M4 Hourly|Covid Death|us_births/D|Saugeen/D|Car Parts|Restaurant|
> |-|-|-|-|-|-|-|-|-|-|-|-|-|-|-|
> |1.1%|1.6%|2.3%|3.6%|5.6%|1.5%|3.8%|1.4%|3.1%|4.7%|19.1%|10.6%|9.1%|0.8%|0.5%|
>
> It is important to note that the MSE/MAE in Table 2 are reported on normalized scales, where small differences can correspond to non-trivial improvements in the raw scale. Judging impact solely by the decimal place can be misleading in this context.
>
> Furthermore, we focus on a broader issue: pre-trained TSFMs often underperform strong classical baselines, limiting their practical adoption. Our pruning-based approach is not just about marginal improvements, but about enabling consistent and reliable adaptation of TSFMs across diverse tasks. The win-rate is used for this purpose.
>
> **Q5 Does the pruning method mostly work on bigger models? The rationale behind choosing those datasets for TTM's expt?**
>
> Pruning is generally more effective on larger models, where activation sparsity and parameter redundancy are more prominent. In our work, we observe that pre-trained TSFMs often exhibit such sparsity, which can be disrupted if all parameters are fine-tuned indiscriminately. Our pruning method is designed to preserve these patterns, enabling effective specialization of TSFMs.
>
> For short time series, TTM automatically selects very small model variants* with limited capacity (e.g., only ~70K parameters for Car Parts). These compact models lack sufficient redundancy. The four additional datasets we selected have longer input sequences, prompting TTM to load larger variants where pruning becomes meaningful and better aligned with our goal of leveraging sparsity.
>
> > *Technical details: Due to the MLP Mixer architecture, there is no unifed TTM to handle varying input lengths. Thus, TTM released many versions pre-trained on time series of different lengths, and its official code selects the appropriate version according to the length of a downstream time series. For example, time series of Car Parts has only 51 time steps, and TTM loads a version named "52-16-ft-l1-r2.1" with only 70,792 parameters. It is reasonable that there are very few redundant parameters in the extremely small model, which is absolutely not suitable for pruning.
>
> **Limitations (missing from the draft)**
>
> We have discussed the limitations in Appendix D, including the limited diversity of evaluation benchmarks and fine-tuning strategies. We have conducted additional experiments during the rebuttal to help address these aspects. Another limitation is that directly pruning the pre-trained model may not fully capture the most effective substructures. A potential improvement is to adopt iterative pruning [1] that prunes models warmed up with a few samples.
>
> > [1] The Lottery Ticket Hypothesis: Finding Sparse, Trainable Neural Networks. ICLR'19.

---

> > ### Comment · Reviewer_SzHa · 2025-08-08
> > **Reply to authors**
> >
> > Thank you for providing the details. While I have more understanding on the proposed method, I still think the evaluation is not exhaustive.
> >
> > - First, If the authors choose to evaluate on GIFT benchmark, selectively picking datasets gives a negative indication, and more importantly, restricts the readers from understanding the true potential of the approach.
> > - Second, the percentage improvement should be provided for all evaluations. The authors said that _"It is important to note that the MSE/MAE in Table 2 are reported on normalized scales, where small differences can correspond to non-trivial improvements in the raw scale"_. In my understanding, relative percentage improvement in the normalized space is employed so that we do not get confused by the raw scale of the data. The **win rate** employed in the draft does not have much significance in regression/forecasting task where the metric is MAE/MSE because the improvement can be at any decimal place. In the above table, the improvement is $< 4\%$ in 10 out of 15 cases.
> >
> > To summarize, while the method is simple and novel, I still think the evaluation data and metrics that are selected should be improved.

---

> ### Author Response · Authors · 2025-08-09
>
> We appreciate the reviewer’s feedback. We would like to clarify our experimental design choices.
>
> Our experimental evaluation on full-shot forecasting performance follows established practices in the TSFM literature [1-7], using the widely adopted ETT, Weather, and Electricity benchmarks. These benchmark have been studied as the standard evaluation protocol in the long-term time series forecasting community over years. Achieving consistent, substantial improvements over baselines remains difficult even for state-of-the-art methods.
>
> In response to the need for broader evaluation, we have expanded beyond the standard benchmarks by incorporating additional datasets spanning diverse domains. To the best of our knowledge, this is among the very first systematic efforts to evaluate full-shot fine-tuning performance across such a variety of datasets in the TSFM literature [1-7].
>
> We believe our experimental evaluation appropriately demonstrates the validity and effectiveness of our approach within the established evaluation framework. The additional dataset diversity we have included represents a meaningful extension beyond standard practice.
>
> > [1] Tiny Time Mixers (TTMs): Fast Pre-trained Models for Enhanced Zero/Few-Shot Forecasting of Multivariate Time Series. NeurIPS'24.
> >
> > [2] Timer-XL: Long-Context Transformers for Unified Time Series Forecasting. ICLR'25.
> >
> > [3] Time-MoE: Billion-Scale Time Series Foundation Models with Mixture of Experts. ICLR'25.
> >
> > [4] A decoder-only foundation model for time-series forecasting. ICML'24.
> >
> > [5] MOMENT: A Family of Open Time-series Foundation Models. ICML'24.
> >
> > [6] Unified Training of Universal Time Series Forecasting Transformers. ICML'24 Oral.
> >
> > [7] VisionTS: Visual Masked Autoencoders Are Free-Lunch Zero-Shot Time Series Forecasters. ICML'25.

---

### Official Review · Reviewer_EirF · 2025-06-30

**Clarity:** 4
**Significance:** 4
**Originality:** 2
**Rating:** 5
**Confidence:** 5

**Summary:**

This paper proposes a pruning and fine-tuning strategy for Time Series Foundation Models (TSFMs). The authors argue that directly fine-tuning pretrained TSFMs often leads to suboptimal performance and overfitting, frequently falling short of smaller models such as PatchTST. Through empirical analysis, they observe that a large portion of attention heads and fully connected channels are redundant, with only a sparse subset of weights are significant to forecasting tasks. Based on the importance of parameters, the model can be pruned and finetuned with only significant parameters.  Experimental results demonstrate that the pruned models consistently outperform those that are directly fine-tuned.

**Questions:**

1. The authors define the **average relative output norm** to reflect the importance of an attention head. Is this importance metric originally proposed? This metric leads to 2 questions:
* Why does the norm of output embedding $\mathbf{o}$ reduce significantly (generally < 5-10 %) compared to that of input embedding $\mathbf{x}$?
* Can we interpret this metric as an indicator of the model’s forecasting capability on a given dataset? If so, why does the performance of large TSFM variants have a better zero-shot performance even they have a smaller proportion of significant heads compared to their smaller variants?

2. For the results in Table 2, except for Chronos and TTM, the winning rates of other TSFMs are relatively low. Could the authors explain why fine-tuning these models fails to outperform PatchTST?

3. For Table 2, it would be helpful to include the zero-shot performance, which can help to analyze how much improvement is achieved via fine-tuning. If space is limited, it could be provided in Appendix.

4. In Figure 4, Moirai_large shows an extremely high proportion of pruned-out parameters, with both input and output channels in deeper layers reaching nearly 100%. In such cases, does the forward pass rely solely on the residual connection in these layers?

5. Since overfitting is identified as a major issue, how does fine-tuning with dropout affect the performance? Have the authors conducted any comparisons or ablation studies in this regard?

6. How one epoch is defined in the experiments? Is it a fixed number of batches, e.g. 100, or a full iteration of all the training samples?

7. The authors use an extremely large batch size (8192) for pruning. Does batch size have a significant effect on the pruning results?

Some errors:
* Typos in the caption of Fig 2 (a)
* The TopK operation in Algorithm 1 should select $s$ with the lowest importance. It appears that the current version selects the ones with the highest importance.

**Ethical Concerns:**

["NO or VERY MINOR ethics concerns only"]

**Final Justification:**

I have carefully reviewed the manuscript and the rebuttal. All of my concerns have been clearly and satisfactorily addressed. This paper tackles a timely and practically important problem, i.e. fine-tuning time series foundation models (TSFMs) effectively and efficiently. While the methodological novelty is not particularly groundbreaking, the application of pruning to TSFMs is exploratory and appears to be practically useful. The experimental evaluation is comprehensive and robust, especially with the inclusion of additional results on more datasets. Overall, I consider this a high-quality paper in the time series forecasting domain, and I will maintain my original scores.

**Limitations:**

Yes, the authors discuss the limitations of their work. However, one important aspect they overlook is the training efficiency of the proposed method. While they show the inference efficiency, it is also crucial to understand how the pruning process impacts training efficiency.

**Paper Formatting Concerns:**

No such issues.

**Quality:**

3

**Strengths And Weaknesses:**

### Strength
1. The paper is well-written and easy to follow, with a clear structure and precise mathematical expressions.
2. The proposed method is simple to understand and general to all TSFMs.
3. Efficient fine-tuning of TSFMs is a timely and important direction to explore, with a great significance in real-world applications.
4. The experiments are comprehensively conducted. The authors validate the proposed method on various TSFMs.

### Weakness
1. The originality of the proposed method seems not very significant. It appears to apply an established method from LLM-pruner to calculate importance of parameters and prune the insignificant ones.
2. Some analysis could be more in-depth. Please refer to the Questions section for further details.


[1] LLM-pruner: on the structural pruning 356 of large language models, NeurIPS 2023

---

> ### Author Rebuttal · Authors · 2025-07-31
>
> **Q1.1: About the average relative output norm. Why does the output norm reduce significantly compared to the input embedding.**
>
> Our inspiration comes from Figure 5c in [1]. Due to residual connections (as formulated in our Eq. 3 and Eq. 1), the relatively small norm of head outputs does not mean significant reduction on the results, since each head output is added into the input hidden states. We define the relative norm to quantify the significance of attention heads when modifying the input hidden states. The small output norm of most attention heads implies that not all heads are necessary in forecasting a specific downstream time series. As it is well accepted that multi-head attention can model heterogenous patterns, we assume that each attention head may play distinct role in handling time series hidden states from a specific domain.
>
> > [1] Deja Vu: Contextual Sparsity for Efficient LLMs at Inference Time. ICML'23.
>
> **Q1.2 Why does large TSFM variants have better zero-shot performance even with a smaller proportion of significant heads compared to smaller variants? **
>
> Though Moirai-large has a smaller proportion of significant heads, the absolute number of significant ones may be greater than Moirai-base, since Moirai-large has 384 heads in total and Moirai-base has 144 heads in total. Increasing model size during pretraining enhances the capacity of time series foundation models to memorize patterns more effectively, which in turn improves their zero-shot forecasting performance.
>
> **Q2: Why fine-tuning Moirai and Timer fails to outperform PatchTST?**
>
> Fine-tuning Moirai and Timer fails to outperform PatchTST because these foundation models already lag behind in zero-shot performance, as shown on the GIFT-Eval leaderboard. Compared to other state-of-the-art time series foundation models, Moirai and Timer exhibit a larger performance gap relative to PatchTST, making it more difficult to close this gap through fine-tuning alone.
>
> **Q3: include zero-shot performance in table.**
>
> Thanks for your advice! We will include it in our future manuscript. You can also refer to Figure 11-14 in our supplementary materials for a brief comparison.
>
> **Q4: Moirai_large shows an extremely high proportion of pruned-out parameters. Does the forward pass rely solely on the residual connection in these layers?**
>
> Thanks for your valuable question! When most channels in the deeper layers are pruned, these layers have minimal impact on the hidden states, meaning that the forward pass primarily relies on the residual connections to carry information forward. As a result, the final output representation is largely determined by the shallower layers, which appear sufficient for accurate prediction. As it is an interesting phenomenon that is also observed in LLMs, we may also apply early-exit techniques [2] to TSFMs, which are promising to reduce computation costs.
>
> > [2] EE-LLM: Large-Scale Training and Inference of Early-Exit Large Language Models with 3D Parallelism.
>
> **Q5: How does fine-tuning with dropout affect the performance?**
>
> Thanks for your suggestions! We compare the average MSE and MAE of Chronos-bolt-base without and with dropout in the following table.
>
> | Methods  | Fine-Tune | (dropout=0) | Fine-Tune | (dropout=0.1) | Prune-then- | Finetune  |
> | -------- | --------- | ----------- | --------- | ------------- | ----------- | --------- |
> | Datasets | MSE       | MAE         | MSE       | MAE           | MSE         | MAE       |
> | ETTh1    | 0.420     | **0.410**   | 0.422     | 0.411         | **0.417**   | **0.410** |
> | ETTh2    | **0.339** | **0.368**   | 0.343     | 0.369         | **0.339**   | **0.368** |
> | ETTm1    | 0.345     | 0.357       | 0.352     | 0.361         | **0.344**   | **0.356** |
> | ETTm2    | 0.255     | 0.297       | 0.263     | 0.300         | **0.247**   | **0.296** |
> | Weather  | 0.261     | 0.268       | 0.268     | 0.278         | **0.215**   | **0.243** |
> | ECL      | 0.154     | 0.240       | 0.153     | 0.239         | **0.150**   | **0.237** |
>
> The results demonstrate that dropout can result in negative effects. We assume that the dropout technique is not suitable for time series. In computer vision tasks, it is naturally required and relatively easy to conduct classification or reconstruction even with noisy pixels. By contrast, it is not suitable to always consider all latent features in time series, which may contain noisy fluctuations or outdated information.
>
> **Q6: How one epoch is defined in the experiments?**
>
> We define one epoch as a full iteration of all the training samples.
>
> **Q7: Does batch size have a significant effect?**
>
> Intuitively, a very large batch size (e.g., approaching the full data size) can capture the comprehensive patterns of the dataset, while smaller batch sizes can increase risks of mistakenly pruning importance channels. We use the batch size to control the iteration numbers and strike a balance between pruning efficiency and effectiveness.

---

> > ### Comment · Reviewer_EirF · 2025-08-01
> >
> > Thank the authors for the response. Most of the questions have been clearly addressed. The results presented in Q5 are particularly interesting. While both dropout and prune-then-finetune are strategies aimed at mitigating overfitting, dropout leads to performance degradation, whereas prune-then-finetune consistently yields positive results. Could the authors further compare and elaborate on the fundamental differences between these two strategies?

---

> ### Author Response · Authors · 2025-08-01
>
> Thanks for your positive evaluation!
>
> When applying dropout during training, it is standard practice to scale the remaining values to preserve the expected mean value. However, this process changes the variance, and after passing through non-linear transformations, the expected mean can differ from that of a model trained without dropout. During training, the final projection layer adapts to the training-time statistics. At inference time, dropout is disabled, resulting in a mismatch between training and inference statistics, which differs from pruning.
> In classification tasks utilizing softmax, the model primarily focuses on relative relationships between probabilities, making it relatively insensitive to statistical discrepancies in the logits. In contrast, regression tasks are predicting absolute quantities and more sensitive to changes in the distribution of latent features. As a result, dropout does not always yield performance improvements in regression settings [1].
> > [1] Effect of Dropout Layer on Classical Regression Tasks. SIU 2020.
>
> Dropout can be beneficial when training models from scratch over multiple epochs, as dropout across repeated samples enhances statistical diversity and improves generalization to various data distributions. However, pre-trained foundation models are effective at cold starting and need only one (or a few) epochs for fine-tuning. It would be better to preserve all data information to help quickly close mismatch between pre-trained knowledge and downstream data patterns.

---

### Official Review · Reviewer_ujDP · 2025-07-02

**Clarity:** 3
**Significance:** 4
**Originality:** 3
**Rating:** 5
**Confidence:** 4

**Summary:**

The authors investigate why large pre-trained Time Series Foundation Models (TSFMs) often fail to outperform smaller task-specific models on downstream forecasting tasks, and propose a prune-then-finetune approach to help address this gap. They leverage the inherent sparsity and redundant components in a pre-trained TSFM by structured pruning (removing less-important channels and attention heads) before fine-tuning on a target dataset. This focuses adaptation on a compact, task-relevant subnetwork, aiming to preserve valuable pre-trained representations while avoiding overfitting the full model. Experiments across six forecasting benchmarks, and 7 recent TSFMs using the prune-then-finetune approach show improved forecasting performance, while also achieving up to 7 times inference speed-ups. The work positions structured pruning not merely as a compression tool but as an effective regularization approach.

**Questions:**

1. Generalizability to Other Datasets: Can the authors comment on the performance of their approach on additional benchmarks beyond those reported, like healthcare? The paper uses six benchmarks, which are fairly standard among time series forecasting papers. Demonstrating the method on a more diverse set (especially where characteristics differ, like highly non-stationary, noisy or shorter series) would strengthen confidence that the approach generalizes widely.

2. Choice of Pruning Ratio and Strategy: How sensitive are the results to the chosen pruning fraction and strategy? How would the model perform under extreme pruning? For example, pruning 90% of the model weights?

3. Effect of Pre-training Quality: The authors assume that the pre-trained TSFMs used have learned useful representations for the task. What about the case where the assumption fails, like when the TSFM exhibits poor zero-shot performance on the target domain? While the authors touch on this in the limitations, it would be interesting to see a concrete experiment where a TSFM pre-trained on electricity data and fine-tuned (with v/s without pruning) on a very different domain (say financial time series). Would pruning still help if the model’s pre-training knowledge is not directly relevant?

**Ethical Concerns:**

["NO or VERY MINOR ethics concerns only"]

**Final Justification:**

I have increased my score from 4 to 5. The authors answered all questions with strong empirical results, and systematically tackled the weaknesses I mentioned in their rebuttal. The methodology is strong, while a little derivative, and the experiments show strong empirical performance. The authors cover a wide range of experiments and justify design decisions empirically too.

**Limitations:**

Yes.

**Paper Formatting Concerns:**

No formatting concerns. Paper is well written, with good formatting.

**Quality:**

3

**Strengths And Weaknesses:**

Strengths:

1. Well-written and motivated paper: The paper is well-written, detailed, with a proper flow motivating the hypothesis and methodology, and then substantiating it with the results.

2. Strong Empirical Evidence: Extensive experiments on 7 TSFMs and 6 benchmarks show the efficacy of the proposed approach.

3. Method is Simple & Model-Agnostic: The authors show that the method works across encoder-only, decoder-only, and MoE TSFMs without architectural changes, thus making it easy to adopt.

4. Performance and Efficiency: The approach benefits both performance and efficiency simultaneously.

Weaknesses:

1. Incremental Novelty: Their pruning metric adapts from LLM-Pruner, and the overall novelty lies more in application and context than in approach itself.

2. Benchmark Diversity: Core results focus on ETT, Weather, Electricity, which aren't sufficient to fully justify the generalizability claims. Experiments on more diverse set of datasets would rectify this weakness.

3. Hyper-parameter Sensitivity: The paper provides limited ablation on pruning ratio and pruning method, thus calling into question how robust performance gains are to these choices.

4. The paper lacks other simple baselines like layer-freezing, adapters, or distillation that could isolate the benefit of removing parameters versus just not updating them.

Authors, please note: I am very open to increasing my score if these above weaknesses are answered/rectified.

---

> ### Author Rebuttal · Authors · 2025-07-31
>
> **Q1 & Q3 & W2: Generalizability to Other Datasets in the financial and healthcare domains. Would pruning still help if the model’s pre-training knowledge is not directly relevant?**
>
> Thanks for your advice! As for healthcare, we additionally perform short-term forecasting over the Covid Death benchmark borrowed from [GIFT-EVAL](huggingface.co/datasets/Salesforce/GiftEval/). As for financial time series, we perform short-term forecasting over the M4 Yearly benchmark. It is worth mentioning that Chronos included merely no *yearly* financial time series, and thus M4 Yearly is distinct to its pretraining knowledge.
>
> The results of Chronos-bolt-base are reported in the following table. Our proposed method outperforms the fine-tuning baseline on almost all the metrics and datasets, and the improvement on Covid Death is more significant than M4 Yearly.
>
> |                                          | Chronos-bolt-base   | MSE         | MAE     | MASE      | CRPS      | NRMSE     |
> | ---------------------------------------- | ------------------- | ----------- | ------- | --------- | --------- | --------- |
> | M4 Yearly (context length$\in[19, 284]$) | Zero-shot           | 3926423     | 943     | 3.507     | 0.121     | 0.318     |
> |                                          | Fine-Tune           | 3226941     | 833     | 3.089     | 0.107     | 0.309     |
> |                                          | Prune-then-Finetune | **3125398** | **816** | **3.007** | **0.104** | **0.283** |
> | Covid Death (context length=212)         | Zero-shot           | 641888      | 164     | 38.9      | 0.047     | 0.301     |
> |                                          | Fine-Tune           | 577110      | 157     | **37.5**  | 0.046     | 0.286     |
> |                                          | Prune-then-Finetune | **239526**  | **93**  | 49.8      | **0.034** | **0.184** |
>
> As for the MLP-based TTM model, its released version suitable for M4 Yearly only has 935 parameters. Applying pruning to this tiny model, we observe a performance decline on the validation set, i.e., the optimal pruning ratio after hyperparameter selection is 0. Thus, we only report the performance of TTM on Covid Deaths as follows:
>
> |                                 | TTM                 | MSE        | MAE     | MASE     | CRPS      | NRMSE     |
> | ------------------------------- | ------------------- | ---------- | ------- | -------- | --------- | --------- |
> | Covid Death (context length=90) | Zero-shot           | 2181495    | 327     | 53.5     | 0.123     | 0.56      |
> |                                 | Fine-Tune           | 688993     | 125     | 30.8     | 0.047     | 0.361     |
> |                                 | Prune-then-Finetune | **584696** | **111** | **29.2** | **0.042** | **0.287** |
>
> From the results, we can see that pruning still brings consistent benefits even when the model's pre-training knowledge is not directly relevant. The sparsification effect essentially acts as an implicit regularizer, helping the model adapt more effectively to new domains with limited or different characteristics from pretraining data.
>
> **Q2 & W3: Sensitivity to the chosen pruning fraction and strategy? Performance under extreme pruning, e.g., pruning 90%?**
>
> That is a valuable question! We studied only moderate pruning ratios (5%-20% per epoch) in our previous hyperparameter tuning and now extend the pruning ratio to 90% according to your advice. In the following table,  we report the average MSE of Chronos-bolt-base with the horizon in {96, 192, 336, 720}.
>
> | Pruning ratio | 5%        | 10%       | 15%   | 20%   | 90%       | Zero-shot |
> | ------------- | --------- | --------- | ----- | ----- | --------- | --------- |
> | ETTh1         | **0.417** | 0.419     | 0.421 | 0.424 | 0.457     | 0.439     |
> | ETTh2         | 0.342     | **0.339** | 0.340 | 0.345 | 0.362     | 0.361     |
> | ETTm1         | 0.347     | 0.348     | 0.347 | 0.348 | **0.344** | 0.391     |
> | ETTm2         | 0.256     | 0.255     | 0.254 | 0.254 | **0.247** | 0.283     |
> | Weather       | 0.226     | 0.220     | 0.220 | 0.221 | **0.215** | 0.242     |
> | Electricity   | 0.151     | 0.150     | 0.150 | 0.151 | **0.149** | 0.154     |
>
> The results are not much sensitive to small pruning ratios. Performance after extreme pruning cannot get recovered after finetuning on small datasets (ETTh1 and ETTh2), where we can observe worse performance than zero-shot forecasting. However, given abundant training samples in other datasets, finetuning the model with 90% parameters pruned (from 207M to 30M) achieves the best performance. This is an interesting phenomenon that can also be observed at Moirai-large on Electricity (see Table 2 in our paper).
>
> As for the pruning strategy, please refer to Appendix C.4 in our supplemental materials. The optimal pruning strategy siginificantly varies by models and datasets, while both pruning strategies outperforms the fine-tuning baseline in most cases.
>
>
>
> **W4: Baselines that just not updating weights**
>
> Thanks for your insightful advice! We implement the classic head probing baseline that only fine-tunes the final projection head and freezes all other layers. As for parameter-efficient fine-tuning, we report results of LoRA in Appendix C.3 in our supplementary materials, where full fine-tuning often outperforms LoRA though insignificantly. In the following table, we compare the average MSE and MAE with the horizon in {96, 192, 336, 720}.
>
> Chronos-bolt-base:
>
> | Methods  | Probing |       | LoRA  |       | Full      | Finetune  | Prune-then- | Finetune  |
> | -------- | ------- | ----- | ----- | ----- | --------- | --------- | ----------- | --------- |
> | Datasets | MSE     | MAE   | MSE   | MAE   | MSE       | MAE       | MSE         | MAE       |
> | ETTh1    | 0.420   | 0.411 | 0.424 | 0.411 | 0.420     | **0.410** | **0.417**   | **0.410** |
> | ETTh2    | 0.348   | 0.373 | 0.344 | 0.370 | **0.339** | **0.368** | **0.339**   | **0.368** |
> | ETTm1    | 0.356   | 0.360 | 0.353 | 0.359 | 0.345     | 0.357     | **0.344**   | **0.356** |
> | ETTm2    | 0.252   | 0.301 | 0.265 | 0.302 | 0.255     | 0.297     | **0.247**   | **0.296** |
> | Weather  | 0.226   | 0.250 | 0.232 | 0.255 | 0.261     | 0.268     | **0.215**   | **0.243** |
> | ECL      | 0.152   | 0.238 | 0.154 | 0.240 | 0.154     | 0.240     | **0.150**   | **0.237** |
>
> TTM:
>
> | Methods  | Probing |       | LoRA  |       | Full  | Finetune | Prune-then- | Finetune  |
> | -------- | ------- | ----- | ----- | ----- | ----- | -------- | ----------- | --------- |
> | Datasets | MSE     | MAE   | MSE   | MAE   | MSE   | MAE      | MSE         | MAE       |
> | ETTh1    | 0.403   | 0.424 | 0.403 | 0.430 | 0.402 | 0.427    | **0.399**   | **0.423** |
> | ETTh2    | 0.333   | 0.385 | 0.333 | 0.385 | 0.333 | 0.385    | 0.333       | **0.383** |
> | ETTm1    | 0.353   | 0.377 | 0.345 | 0.377 | 0.348 | 0.379    | **0.342**   | **0.373** |
> | ETTm2    | 0.252   | 0.307 | 0.244 | 0.310 | 0.247 | 0.311    | **0.245**   | **0.306** |
> | Weather  | 0.225   | 0.264 | 0.222 | 0.264 | 0.223 | 0.265    | **0.221**   | **0.262** |
> | ECL      | 0.162   | 0.261 | 0.192 | 0.287 | 0.161 | 0.258    | **0.160**   | **0.256** |
>
> The results demonstrate that full finetuning is a strong baseline, and neither head probing nor LoRA exhibit significant performance improvement. In a few cases, head probing or LoRA outperforms full finetuning, while our proposed prune-then-finetune method still achieves the best performance.

---

> > ### Comment · Reviewer_ujDP · 2025-08-06
> > **Strong Rebuttal, primary concerns addressed**
> >
> > Thank you for your detailed and systematic rebuttal. I am happy that my primary concerns with the paper have been addressed with strong empirical results and relevant experiments. I am increasing my score for the paper to 5, as I believe that it is appropriate, given the depth of experiments performed.

---

> > > ### Author Response · Authors · 2025-08-07
> > >
> > > Thanks for your positive recommendation! We will improve our future manuscript according to your suggestions :)

---

> ### Author Response · Authors · 2025-08-02
>
> **Q1 & Q3 & W2: Generalizability to Other Datasets**
>
> We have just accomplished experiments on more time series that are much short and non-stationary. We would like to share our new results in the following tables, where the reported MSE and MAE are based on raw values without z-score normalization.
>
> | | Chronos-bolt-base | MSE | MAE | MASE | CRPS | NRMSE |
> | -- | --- | --- | -- | - | - | - |
> |M4 Quarterly|Zero-shot|1878597|578|1.224|0.077|0.229|
> ||Fine-Tune|1775734|571|**1.221**|0.075|0.223|
> ||Prune-then-Finetune|**1749123**|**551**|1.225|**0.074**|**0.221**|
> |M4 Monthly|Zero-shot|1934416|565|0.949|0.094|0.289|
> ||Fine-Tune|1894232|571|0.946|0.095|0.286|
> ||Prune-then-Finetune|**1820581**|**556**|**0.920**|**0.091**|**0.280**|
> | M4 Hourly (length$\in [78, 1008]$) | Zero-shot | 1154587 | 244 | 0.837 | 0.025 | 0.147 |
> | | Fine-Tune | 1032867 | 239 | 0.943 | 0.026 | 0.139 |
> | | Prune-then-Finetune | **937494** | **227** | **0.805** | **0.024** | **0.131** |
> | US Births/D (healthcare, length=2048) | Zero-shot | 261850 | 329 | 0.485 | 0.026 | 0.048 |
> | | Fine-Tune | 171933 | 248 | 0.365 | 0.019 | 0.039 |
> | | Prune-then-Finetune | **168877** | **241** | **0.354** | **0.018** | **0.039** |
> | Saugeen/D (nature, length=2048) | Zero-shot | 1082 | 12.8 | 2.840 | 0.338 | 1.065 |
> | | Fine-Tune | 1057 | 13.0 | 2.889 | 0.351 | 1.053 |
> | | Prune-then-Finetune | **1052** | **11.9** | **2.642** | **0.328** | **1.051** |
> | Car Parts (sales, length=51) | Zero-shot |1.39| 0.48  | 0.855 | 0.995 | 2.827 |
> | | Fine-Tune | 1.378 | 0.438 | 0.835 | 0.916 | 2.815 |
> | | Prune-then-Finetune |**1.365** | **0.435** | **0.826** | **0.910** | **2.802** |
> | Restaurant (sales, length$\in [67, 478]$) | Zero-shot |146.6|7.329| 0.700 | 0.264 | 0.557 |
> | | Fine-Tune | 143.8 |7.192| 0.687 | 0.259 | 0.552 |
> | | Prune-then-Finetune |**142.5**|**7.159** |**0.684**|**0.258**|**0.549** |
>
> From the results, we can see that our proposed method generalize well on these diverse datasets and outperforms the finetuning baseline.
>
> **W1: The pruning metric adapts from LLM-Pruner**
>
> Thanks for your insightful comment! Actually, LLM-Pruner borrows the pruning metric from previous works [2] in the computer vision field. Despite the simplicity of its design, the significance of LLM-Pruner lies in that it is the first work that demonstrates the efficiency benefits of structured pruning for LLMs. We are also the first to explore pruning over time series models, while the difference is our particular focus on improving forecasting performance. Our work validates the effectiveness of pruning with only minimal and straightforward designs, overturning the conventional assumption of lossy compression in other fields. We believe that this would inspire future works to develop more advanced pruning techniques for higher predictive performance. Additionally, we present novel empirical studies that discover the sparsity and redundancy in time series foundation models. It would offer insights into why both dense models and sparse mixture-of-experts models can handle heterogeneous time series.
>
> > [2] Group Fisher Pruning for Practical Network Compression. ICML'21.

---

### Official Review · Reviewer_3GcC · 2025-07-03

**Clarity:** 3
**Significance:** 3
**Originality:** 2
**Rating:** 5
**Confidence:** 3

**Summary:**

The paper proposes a prune-then-finetune approach for adapting pre-trained large time-series forecasting models. They prune the TSFM first by removing redundant channels/heads guided by a loss-based importance score, and then fine-tune the smaller, specialized subnetwork on the downstream data. Empirical results show that pruning + fine-tuning reduces error significantly vs fine-tuning alone, and faster inference.

**Questions:**

Did you explore how sensitive your pruning decisions and final forecasting performance are to different random seeds?
Is the set of pruned channels largely the same across them, or do you see very different subnetworks?
It might be interesting to visualize the difference, perhaps for only a small subset of the complete set of experiments if the resources allow for it.

**Ethical Concerns:**

["NO or VERY MINOR ethics concerns only"]

**Limitations:**

yes

**Paper Formatting Concerns:**

No formatting concerns

**Quality:**

3

**Strengths And Weaknesses:**

Strengths:
1. The problem of adapting a large TSFM is very important in the time-series field, and the proposed approach is promising.
2. Both the forecasting performance gains and computation efficiency improvements are significant
3. The overall approach is simple, and in principle applicable to other domains.
4. The paper is well written, and the experiments are adequate

Weaknesses:
1. No repeated trials for statistical significance analysis, however that is acceptable for large-scale models like these
2. It would be interesting to see if the same approach results in similar improvements for tasks such as classification, imputation, and anomaly detection using a foundation model such as MOMENT [1].

[1] Goswami, M., Szafer, K., Choudhry, A., Cai, Y., Li, S., & Dubrawski, A. (2024). Moment: A family of open time-series foundation models. arXiv preprint arXiv:2402.03885.

---

> ### Author Rebuttal · Authors · 2025-07-31
>
> **Q1: Sensitivity to random seeds**
>
> Thanks for your advice! We prune Chronos-bolt-base in one epoch with two different random seeds, and the Jaccard similarity between the two sets of pruned channels is reported in the following table.
>
> |                    | ETTh1 | ETTh2 | ETTm1 | ETTm2 | Weather |
> | ------------------ | ----- | ----- | ----- | ----- | ------- |
> | Jaccard similarity | 97.6% | 98.3% | 95.1% | 94.7% | 93.8%   |
>
> We do not observe significantly different subnetworks. One reason is that, during pruning, we set a large batch size (e.g., 8192 as recommended) to ensure that each sampled batch could cover diverse data patterns. Consequently, randomness mostly lies in the order of training batches, of which the total number is small (e.g., ETTh1 has 14 batches).
>
> We will visualize the layer-wise and channel-wise difference by figures in our future manuscript.
>
> **W1: Repeated trials**
>
> We repeat runs with 5 different random seeds, and we report MSE with standard deviations in the following table.
>
> |                                 | ETTh1          | ETTh2          | ETTm1          | ETTm2          | Weather        |
> | ------------------------------- | -------------- | -------------- | -------------- | -------------- | -------------- |
> | Chronos-bolt-base (horizion=96) | 0.366 ± 0.0005 | 0.416 ± 0.0007 | 0.282 ± 0.0011 | 0.157 ± 0.0009 | 0.142 ± 0.0015 |
>
> The standard deviations are considerably small. We posit that fine-tuning foundation models have fewer randomness than traditional models since all trials are based on the same pretrained weights without random initialization.
>
> **W2:  Other tasks using MOMENT**
>
> Due to the time limit, we are preparing codes and experiments, and we may provide results in the next week. Thanks!

---

> > ### Comment · Area_Chair_Q1zq · 2025-08-06
> >
> > Dear reviewers,
> >
> > Kindly consider revisiting the rebuttal and providing any updated thoughts or clarifications if needed. Your feedback will be valuable for the final decision.
> >
> > Best Regards
> >
> > AC

---

> ### Author Response · Authors · 2025-08-08
> **W2: Imputation task using MOMENT**
>
> **W2: Imputation task using MOMENT**
>
> Thanks for your interest! We have performed imputation tasks with MOMENT fine-tuned by different methods. The sequence length is 512, and each patch has a 25% probability to be masked. As shown in the following table, our approach results in similar improvements. We would like to study anomaly detection tasks as a future work. Thanks for your valuable advice!
>
> |  | MOMENT | (Finetune) | MOMENT | (Ours) |
> |---|---|---|---|---|
> |  | MSE | MAE | MSE | MAE |
> | Weather | 0.032 | 0.072 | **0.029** | **0.068** |
> | ETTh1 | 0.140 | 0.236 | **0.138** | **0.233** |
> | ETTh2 | 0.067 | 0.165 | **0.065** | **0.162** |
> | ETTm1 | 0.068 | 0.165 | **0.064** | **0.161** |
> | ETTm2 | 0.027 | 0.101 | **0.025** | **0.096** |
> | Electricity | 0.091 | 0.208 | **0.089** | **0.205** |

---

### Note · Authors · 2025-08-14

Many thanks to the reviewers and the ACs for the dedication to reviewing our paper!

We are devoted to specializing large TSFMs for state-of-the-art downstream performance, which is recognized as a "timely" and "pretty important" problem (Reviewers 3GcC, EirF, SzHa). All reviewers acknowledged that our proposed method is "simple and novel" (Reviewers 3GcC, EirF, ujDP, SzHa). We sincerely hope that the broad community can draw more attention to effectively fine-tuning TSFMs, and new ideas may emerge from the discovered sparsity phenomenon in TSFMs.

During the rebuttal, we further tested our approach on additional datasets spanning various domains, including traffic, healthcare, sales, nature, and finance. The results show that our method generalizes well across 15 datasets with an average error reduction of 4.9% compared to the baseline. The depth of these experiments has been acknowledged by Reviewers EirF & ujDP, and our empirical results have been considered "adequate" (Reviewer 3GcC), "extensive" (Reviewer ujDP), and "strong" (Reviewers EirF & ujDP) enough to address concerns about the diversity of the evaluation.

---

### Decision · Program_Chairs · 2025-09-17

**Decision:**

Accept (poster)

**Comment:**

This paper addresses the important and timely problem of efficiently adapting large pre-trained TSFMs to downstream forecasting tasks. The authors propose a prune-then-finetune framework, which removes redundant channels and attention heads based on a loss-based importance score, and then fine-tunes the resulting compact subnetwork. This approach is simple, model-agnostic, and applicable to encoder-only, decoder-only, and MoE TSFMs. Experiments across seven TSFMs and six benchmarks show consistent performance improvements and significant inference speed-ups.

Strengths:

1. Tackles a timely and practical problem in TSFM adaptation.

2. Method is simple, broadly applicable, and empirically effective.

3. Demonstrates both performance gains and computational efficiency.

4. Well-written with extensive experimental validation.

Weakness:
1.  Algorithmic sensitivity & hyperparameters: The pruning-then-finetune method requires gradually expanding the scope of pruned components and accumulating sufficient feedback over batches to stabilize the importance score. These design choices introduce multiple hyperparameters and can make the algorithm potentially unstable and difficult to use reliably in practical deployments.

2.  Benchmark diversity: Experiments are concentrated on ETT, Weather, and Electricity; generalizability to other datasets (e.g., traffic) remains unclear.

The work is solid, well-motivated, and demonstrates promising empirical results in adapting large TSFMs using a prune-then-finetune approach. Although questions remain regarding novelty, robustness, hyperparameter sensitivity, and generalization to broader benchmarks, the method offers a clear and valuable practical contribution to the community. Considering these factors, I recommend acceptance.